# Fast Routing under Uncertainty: Adaptive Learning in Congestion Games with Exponential Weights

**Dong Quan Vu**          **Kimon Antonakopoulos**
Univ. Grenoble Alpes, CNRS, Inria, Grenoble INP, LIG 38000, Grenoble, France
dong-quan.vu@inria.fr    kimon.antonakopoulos@inria.fr

**Panayotis Mertikopoulos**
Univ. Grenoble Alpes, CNRS, Inria, Grenoble INP, LIG 38000, Grenoble, France
Criteo AI Lab
panayotis.mertikopoulos@imag.fr

## Abstract

We examine an adaptive learning framework for nonatomic congestion games where the players' cost functions may be subject to exogenous fluctuations (e.g., due to disturbances in the network, variations in the traffic going through a link, etc.). In this setting, the popular multiplicative / exponential weights algorithm enjoys an $\mathcal{O}(1/\sqrt{T})$ equilibrium convergence rate; however, this rate is suboptimal in *static* environments – i.e., when the network is not subject to randomness. In this static regime, accelerated algorithms achieve an $\mathcal{O}(1/T^2)$ convergence speed, but they fail to converge altogether in *stochastic* problems. To fill this gap, we propose a novel, *adaptive exponential weights* method – dubbed ADAWEIGHT – that seamlessly interpolates between the $\mathcal{O}(1/T^2)$ and $\mathcal{O}(1/\sqrt{T})$ rates in static and stochastic environments respectively. Importantly, this "best-of-both-worlds" guarantee does not require *any* prior knowledge of the problem's parameters or any tuning by the optimizer; in addition, the method's convergence speed depends subquadratically on the size of the network (number of vertices and edges), so it scales gracefully to large, real-life urban networks.

## 1  Introduction

Navigation apps like Google Maps and Waze have user bases numbering in the hundreds of millions, and they may receive upwards of $10^4$ routing requests per second; in fact, Google Maps alone exceeded one billion monthly active users in 2019, and its interface routinely receives up to $10^5$ requests during rush hour in major metropolitan centers [10]. This vast number of users must be routed efficiently, in real-time, and without causing any "ex-post" regret at the user end; otherwise, if a user could have experienced better travel times along a non-recommended route, they would have no incentive to follow the app recommendation in the first place.

In the language of congestion games [40], this requirement is known as a "Wardrop equilibrium", and it is typically represented as a high-dimensional vector describing the traffic flow along each path in the network [49]. Ideally, this equilibrium should be computed *before* making a recommendation, so as to minimize the number of disgruntled users. In practice however, this is rarely possible: the state of the network typically depends on random factors that vary from one epoch to the next (weather conditions, traffic accidents, fluctuations in the total number of commuters in the system, etc.), so it is generally unrealistic to expect that such a recommendation can be made in advance.

Our paper takes a learning approach to this problem: routing recommendations are provided in an online manner, and they are subsequently updated "on the fly" once the state of the network has

been observed. In more detail, motivated by applications to GPS routing and navigation apps, we consider an adaptive recommendation paradigm that unfolds as follows:

1. At each epoch $t = 1, 2, \ldots$, a centralized control interface – such as Google Maps – determines a routing flow for its users and provides a recommendation accordingly.
2. After making a recommendation, the interface observes the travel times of the network's users; based on this feedback, it updates the routing recommendation and the process repeats.

**Main challenges.** There are several key challenges that arise in this setting. First and foremost, learning methods that are well-suited to rapidly fluctuating environments may be highly suboptimal in static networks and vice versa. Second, the problem's dimensionality – the number $P$ of available paths – is exponential in the size of the underlying network, so it is crucial to propose learning methods that remain efficient in large networks. Finally, methods that require prior knowledge of the problem's parameters – e.g., the smoothness modulus of the network's latency functions – are beyond reach because such knowledge cannot be realistically obtained by the optimizer.

In view of all this, our paper seeks to answer the following question:

*Is it possible to design an adaptive, parameter-agnostic algorithm that is simultaneously optimal in static and stochastic networks, and whose convergence rate is polynomial in the network's size?*

**Our contributions in the context of related work.** Our paper proposes a novel, *adaptive exponential weights* algorithm – dubbed ADAWEIGHT – which enjoys the following desirable properties:

1. In static networks, the method converges to a Wardrop equilibrium at a rate of $\mathcal{O}((\log P)^{\frac{3}{2}}/T^2)$.
2. In stochastic networks, it converges to a mean Wardrop equilibrium at an $\mathcal{O}((\log P)^{\frac{3}{2}}/\sqrt{T})$ rate.
3. These rates are attained without any prior tuning by the optimizer.

In the above, $T$ denotes the learning horizon (number of epochs) and $P$ is the number of paths used to route traffic in the network. Thus, even though $P$ may grow exponentially, the logarithmic dependence on $P$ ensures that the algorithm's runtime remains *polynomial*, and in fact, subquadratic, in the size of the network. To the best of our knowledge, ADAWEIGHT is the first method that simultaneously achieves these desiderata; to provide the necessary context, we give below a detailed account of the related work on the topic.

The *static* regime of our learning model matches the standard framework of Blum et al. [7] who showed that a variant of the classic *exponential weights* (EW) algorithm [4, 5, 31, 48] converges to a Wardrop equilibrium at an $\mathcal{O}(1/\sqrt{T})$ rate (in the Cesàro, time-averaged sense). This result was subsequently extended to *stochastic* congestion games by Krichene et al. [22, 24], who showed that the EW algorithm also enjoys an $\mathcal{O}(1/\sqrt{T})$ convergence rate to *mean* Wardrop equilibria (again, in a Cesàro sense). As we discuss in the sequel, the convergence speed of the EW algorithm of Blum et al. [7] and Krichene et al. [22] is $\mathcal{O}(\log P/\sqrt{T})$ in both cases; however, if the method's learning rate is not chosen appropriately, the EW algorithm may lead to non-convergent, chaotic behavior, even in symmetric congestion games over a 2-link Pigou network [42].

In general equilibrium problems, the $\mathcal{O}(1/\sqrt{T})$ rate cannot be improved (see e.g., [1, 8]) without more stringent assumptions – such as strong monotonicity and the like. However, nonatomic congestion games are well known to admit a convex potential – sometimes referred to as the *Beckmann–McGuire–Winsten* (BMW) potential [6] – so the $\mathcal{O}(1/\sqrt{T})$ convergence guarantee of Blum et al. [7] is *not* optimal in the static regime (i.e., when costs functions do not change overtime). In this static regime, the optimal smooth convex minimization rate is $\mathcal{O}(1/T^2)$ [34, 36], and it is achieved by the seminal "accelerated gradient" algorithm of Nesterov [35]. If applied directly to our problem, Nesterov's algorithm has a catastrophic $\Theta(P)$ dependence on the number of paths; however, by coupling it with a "mirror descent" template in the spirit of [38, 50], Krichene et al. [24] proposed an accelerated method with an exponential projection step that is particularly well-suited for congestion problems. In fact, going a step further, it is possible to design an accelerated exponential weights method – ACCELEWEIGHT for short – that achieves an $\mathcal{O}(\log(P)/T^2)$ rate in static environments. Note also that Nakamura et al. [33] recently shows that an $\mathcal{O}(\exp(-T))$ rate is achievable under a stronger assumption that the BMW potential is strongly convex (which is beyond the scope of consideration of our paper).

| | EW | ACCELEWEIGHT | UNIXGRAD | UPGD | AC-SA | ADAWEIGHT [ours] |
|---|---|---|---|---|---|---|
| STATIC | $\log P/\sqrt{T}$ | $\log P/T^2$ | $P/T^2$ | $\log P/T^2$ | $\log P/T^2$ | $\log P/T^2$ |
| STOCH. | $\log P/\sqrt{T}$ | ✕ | $P/\sqrt{T}$ | ✕ | $\log P/\sqrt{T}$ | $\log P/\sqrt{T}$ |
| ANYTIME | partially | ✓ | ✓ | ✕ | ✕ | ✓ |
| PAR. AGN. | partially | ✕ | ✓ | ✓ | ✕ | ✓ |

**Table 1:** Overview of related work. All rates are reported in the $\mathcal{O}(\cdot)$ sense; "par. agn." means that the method is *parameter-agnostic*, i.e., it does not require prior tuning or knowledge of the problem's parameters.

Crucially, the learning rate parameter of ACCELEWEIGHT must be tuned with prior knowledge of the problem's smoothness parameters; moreover, despite its optimality in the static regime, the method fails to converge altogether in stochastic problems. The universal algorithm of Nesterov [39] provides a method to resolve the former issue, but it relies on a line-search mechanism that cannot be adapted to a stochastic framework, so it does not resolve the latter. On the flip side, the accelerated stochastic approximation (AC-SA) algorithm of Lan [26] achieves optimal rates in both the static and stochastic regimes, but it does not provide anytime guarantees (the iteration budget must be fixed as a function of the accuracy threshold required), and it assumes full knowledge of the smoothness modulus of the game's cost functions (which is not realistically available to the optimizer).

Our approach adapts to the problem's smoothness parameters via an "inverse-sum-of-squares" learning rate in the spirit of ADAGRAD [16] – though ADAGRAD itself lacks an acceleration mechanism, so it is suboptimal in static environments [29, 30]. To the best of our knowledge, the first order-optimal interpolation result was achieved by the ACCELEGRAD algorithm of Levy et al. [29] for *unconstrained* problems (and assuming knowledge of a compact set containing a solution of the problem). The UNIXGRAD proposal of Kavis et al. [20] subsequently achieved the desired adaptation in constrained problems, but under the requirement of a bounded Bregman diameter. This requirement rules out the EW template (the simplex has infinite entropic diameter), so the convergence speed of UNIXGRAD is polynomial in the number of paths, and hence unsuitable for large network instances. For convenience, we compare all these works in Table 1 above.

## 2 Problem setup

**2.1. The game.** Building on the classical model of Beckmann et al. [6], we will consider a class of routing games defined by three basic primitives: *(i)* the game's *network structure*; *(ii)* the associated set of *traffic demands*; and *(iii)* the network's *cost functions*. The formal definition is as follows:

1. **The network structure:** Consider a directed graph $\mathcal{G} \equiv \mathcal{G}(\mathcal{V}, \mathcal{E})$ with vertex set $\mathcal{V}$ and edge set $\mathcal{E}$. The focal point of interest is a set of pre-determined *origin-destination* (O/D) pairs $(O_i, D_i) \in \mathcal{V} \times \mathcal{V}$ indexed by $i \in \mathcal{N} = \{1, \ldots, N\}$. Each pair $i \in \mathcal{N}$ is associated to a *traffic demand* $M_i > 0$ that is to be routed from $O_i$ to $D_i$ via a fixed set of *paths* (or *routes*) $\mathcal{P}_i$ joining $O_i$ to $D_i$ in $\mathcal{G}$. We denote the set of all such paths in the network by $\mathcal{P} := \bigcup_{i \in \mathcal{N}} \mathcal{P}_i$ and, for concision, we denote the corresponding cardinalities as $P_i := |\mathcal{P}_i|$ and $P := |\mathcal{P}| = \sum_i P_i$. We also write $M_{\text{tot}} := \sum_{i \in \mathcal{N}} M_i$ and $M_{\max} := \max_{i \in \mathcal{N}} M_i$ for the total and maximum traffic demand associated to network's O/D pairs.

2. **Routing flows:** In order to route the traffic, the set of feasible *flow profiles* is defined as

$$\mathcal{X} := \left\{ x \in \mathbb{R}_+^{\mathcal{P}} : \sum_{p \in \mathcal{P}_i} x_{i,p} = M_i, i = 1, \ldots, N \right\} \tag{1}$$

i.e., as the product of scaled simplices $\mathcal{X} = \prod_i M_i \Delta(\mathcal{P}_i)$. In turn, each feasible flow profile $x \in \mathcal{X}$ induces on each edge $e \in \mathcal{E}$ a *routing load* $\mu_e(x) = \sum_{i \in \mathcal{N}} \sum_{p \in \mathcal{P}_i} \mathbb{1}_{\{e \in p_i\}} x_{i,p}$, i.e., the accumulated mass of all traffic associated to the focal set of O/D pairs that goes through $e$.

3. **Congestion cost:** The traffic routed through a given edge $e \in \mathcal{E}$ incurs a *congestion cost* (or *latency*) depending on the total traffic on the edge and/or any other exogenous factors. Formally, we will collectively encode all such factors in a *state variable* $\omega \in \Omega$ taking values in some ambient probability space $(\Omega, \mathcal{F}, \mathbb{P})$. We will further assume that each edge $e \in \mathcal{E}$ is endowed with an *edge-cost function* $c_e : \mathbb{R}_+ \times \Omega \to \mathbb{R}_+$; in this way, given a flow profile $x \in \mathcal{X}$ and a state $\omega \in \Omega$, the cost to traverse edge $e \in \mathcal{E}$ will be $c_e(\mu_e(x), \omega)$. Analogously, the cost to

traverse a path $p \in \mathcal{P}$ will be given by the induced *path-cost function* $c_p \colon \mathcal{X} \times \Omega \to \mathbb{R}_+$ defined as $c_p(x, \omega) = \sum_{e \in p} c_e(\mu_e(x), \omega)$.

The only assumption that we will make for the game's cost functions is as follows:

**Assumption 1.** Each cost function $c_e(\rho, \omega)$, $e \in \mathcal{E}$ is measurable in $\omega$ and non-decreasing, bounded and Lipschitz continuous in $\rho$. Specifically, there exist $H > 0$ and $L > 0$ such that $c_e(\rho, \omega) \leq H$ and $|c_e(\rho, \omega) - c_e(\rho', \omega)| \leq L|\rho' - \rho|$ for all $\rho, \rho' \in [0, M_{\mathrm{tot}}]$ and all $\omega \in \Omega$, $e \in \mathcal{E}$.

*Remark.* Assumption 1 is a very mild regularity condition that is satisfied by most congestion models that occur in practice – including BPR, polynomial, or regularly-varying latency functions, cf. [7, 11, 13, 22, 25, 27, 40, 41, 43, 45] and references therein. For this reason, we will treat Assumption 1 as a standing, blanket assumption and we will not mention it explicitly in the sequel.

The key difference between our problem setup and other nonatomic congestion models in the literature [40, 45] is the advent of uncertainty in the game's cost functions (as modeled by the game's ambient state variable $\omega \in \Omega$). For concreteness, we demonstrate below two examples of this type of uncertainty that could arise in practice:

**Example 1** (Stochastically perturbed BPR costs)**.** Standard BPR models [13, 27, 41, 43], under the form $c_e^{\mathrm{BPR}}(\rho) = a_e + b_e(\rho/\mathrm{cap}_e)^r, \forall \rho \geq 0$, capture the effects of a link's length, its capacity, free-flow speed, etc., on urban network congestion. However, they neglect other miscellaneous factors such as weather conditions, accidents and other random factors which may cause a temporary increase – or decrease – in congestion. In our notation, if $\omega_e$ is an additive random fluctuation for the cost of edge $e \in \mathcal{E}$, the induced cost at a flow $x \in \mathcal{X}$ becomes $c(x, \omega) = c_e^{\mathrm{BPR}}(\mu_e(x)) + \omega_e$. The noise can also appear in cases where the actual latency / congestion can only be measured up to a certain error (a case of vital importance for Internet-like networks). §

**Example 2** (BPR models with exogenous loads)**.** In practice, the total traffic is the aggregation of all commuters, irrespective of whether they are using a navigation app or not. With this in mind, consider the problem of a navigation app making a flow recommendation $x \in \mathcal{X}$ for its users. If $\omega_e$ denotes the *exogenous* traffic load on edge $e \in \mathcal{E}$ (i.e., commuters *not* using the app), the total load on $e$ will be $\mu_e(x) + \omega_e$. Thus, following the BPR model, the induced cost is $c_e(x, \omega) = c_e^{\mathrm{BPR}}(\mu_e(x) + \omega_e)$, i.e., the randomness is woven *implicitly* in the model. §

**2.2. The mean game and equilibrium flows.** In the above framework, each exogenous state variable $\omega \in \Omega$ determines an instance of a routing game, defined formally as a tuple $\Gamma_\omega \equiv \Gamma_\omega(\mathcal{G}, \mathcal{N}, \mathcal{P}, c_\omega)$ where $c_\omega$ is shorthand for the network's cost functions $\{c_e(\cdot, \omega)\}_{e \in \mathcal{E}}$ instantiated at $\omega$. Of course, in analyzing the game, each individual instance $\Gamma_\omega$ is meaningless by itself, unless $\mathbb{P}$ assigns positive probability only to a *single* $\omega$. For this reason, we will instead focus on the *mean game* $\Gamma \equiv \Gamma(\mathcal{G}, \mathcal{N}, \mathcal{P}, C)$ which has the same network and routing flow structure as every $\Gamma_\omega$, $\omega \in \Omega$, but whose congestion costs are determined by the *mean cost functions* $C_p(x) = \mathbb{E}[c_p(x, \omega)]$.

Now, motivated by the route recommendation problem described in the introduction, we will focus on learning *equilibrium flows* where the controller can guarantee Wardrop's principle on average [49], i.e., that *all traffic is routed along a path with minimal mean cost.* Formally, we have:

**Definition 1** (Mean equilibrium flows)**.** We say that $x^* \in \mathcal{X}$ is a *mean equilibrium flow* if and only if, for all $i \in \mathcal{N}$ and all $p, q \in \mathcal{P}_i$ such that $x^*_{i,p} > 0$, we have $C_p(x^*) \leq C_q(x^*)$.

*Remark.* Definition 1 means that, on average, no user has an incentive to deviate from the recommended route; obviously, when the support of $\mathbb{P}$ is a singleton, we recover the usual definition of a *Wardrop equilibrium* [6, 40, 49]. This special case will be particularly important and we describe it in detail in the next section.

Importantly, the problem of finding an equilibrium flow of a (fixed) routing game $\Gamma_\omega$ admits a *potential function* – often referred to as the Beckmann–McGuire–Winsten (BMW) potential [6, 15]. Specifically, for a given instance $\omega \in \Omega$, the BMW potential is defined as

$$F_\omega(x) := \sum_{e \in \mathcal{E}} \int_0^{\mu_e(x)} c_e(\rho, \omega) \, d\rho \quad \text{for all } x \in \mathcal{X}, \tag{BMW}$$

and it has the property that $\arg\min F_\omega$ coincides with the set $\mathrm{Eq}(\Gamma_\omega)$ of equilibrium flows of $\Gamma_\omega$.

In our stochastic context, a natural question that arises is whether the potential property for each *fixed* $\omega \in \Omega$ can be extended to the mean game $\Gamma$ when $\omega$ is randomly generated. Clearly, the most

direct candidate for a potential function in this case is is the averaged BMW potential:

$$F(x) = \mathbb{E}[F_\omega(x)] \coloneqq \mathbb{E}\left[\sum_{e \in \mathcal{E}} \int_0^{\mu_e(x)} c_e(\rho, \omega)\, d\rho\right]. \tag{2}$$

Indeed, as we show in Appendix B, we have:

**Proposition 1.** *A flow profile $x^* \in \mathcal{X}$ is a* mean equilibrium flow *if and only if it is a minimizer of $F$ over $\mathcal{X}$; more succinctly,* $\mathrm{Eq}(\Gamma) = \arg\min_{x \in \mathcal{X}} F(x)$.

In view of the above, $F$ provides a natural merit function for examining how close a given flow profile $x \in \mathcal{X}$ is to being an equilibrium; on that account, *all our convergence certificates in the sequel will be stated in terms of $F$*. Note also that since $c_e, e \in \mathcal{E}$, are continuous and non-decreasing, $F$ is a differentiable, convex function on $\mathcal{X}$. However, since the probability law $\mathbb{P}$ on $\Omega$ is not known, we will *not* assume that $F$ (and/or its gradients) can be explicitly computed in general.

**2.3. The learning model.** The last component of our model is the actual learning process that unfolds over time. The specific sequence of events that we will consider evolves as follows:

1. At each stage $t = 1, 2, \ldots$, the navigation interface selects a flow profile $X^t \in \mathcal{X}$ and makes the corresponding routing recommendation to its users.

2. Concurrently, the state $\omega^t$ of the network is drawn from $\Omega$ (i.i.d. relative to $\mathbb{P}$).

3. The interface observes the realized congestion costs $c_e(\mu_e(X^t), \omega^t)$ along each $e \in \mathcal{E}$ (possibly up to some error); subsequently, the flow recommendation is updated, and the process repeats.

We will refer to this general model as the **stochastic regime**. For concreteness, we discuss below two special cases that have attracted particular interest in the literature:

**Example 3** (Static environments)**.** In the absence of randomness, $\mathbb{P}$ is supported on a single instance $\omega \in \Omega$, so we have $\omega^t = \omega$ for all $t = 1, 2, \ldots$, and we assume that the navigation interface measures directly $C(X^t) = c(X^t, \omega)$. This setup matches the deterministic model of Blum et al. [7], Fischer and Vöcking [17] and Krichene et al. [23, 25], and we will refer to it as the **static regime**. §

**Example 4** (Routing games with noisy observations)**.** Consider the setting where only a stochastic "perturbed" cost $c_e(\mu_e(x), \omega^t) = c_e(\mu_e(x), \omega) + \omega_e^t$ is observed when a flow profile $x \in \mathcal{X}$ is employed at time $t$ (here, $\omega \in \Omega$ is fixed). When $\omega_e^t$ is the random noise such that $\mathbb{E}[\omega_e^t] = 0, \forall e, \forall t$, our learning model is reduced to the routing game with noisy observations studied by Krichene et al. [22, 25]. §

For a given learning window, it might not be a priori clear whether the system is in the static or stochastic regime. As we shall see in Section 3, standard stochastic algorithms are suboptimal in the static regime, while order-optimal deterministic algorithms may fail to converge altogether in the stochastic regime. As such, the key question that we aim to answer is as follows: *is it possible to design routing algorithms that automatically adapt to the appropriate setting and provide optimal convergence guarantees in both static and stochastic environments?* We address this in Section 4.

## 3  Non-adaptive methods

To set the stage for the analysis to come, we begin by presenting the equilibrium convergence properties of two learning methods that are tailor-made for each of the two basic regimes described in the previous section: the "vanilla" exponential weights algorithm for the stochastic case, and an accelerated exponential weights method for the static one. Both algorithms rely crucially on the (rescaled) *logit choice map* $\Lambda \colon \mathbb{R}^P \to \mathcal{X}$, given in components as

$$\Lambda_p(Y) = \frac{M_i \exp(Y_{i,p})}{\sum_{q \in \mathcal{P}_i} \exp(Y_{i,q})} \quad \text{for all } p \in \mathcal{P}_i \text{ and all } i \in \mathcal{N}. \tag{3}$$

For concision, we will also write $C_p^t$ for the total congestion cost measured for path $p \in \mathcal{P}$ at stage $t$, and $C^t = (C_p^t)_{p \in \mathcal{P}}$ for the profile thereof.

**3.1. Exponential weights in the stochastic regime.** We begin by presenting the standard exponential weights algorithm in the stochastic regime. In pseudocode form, we have:

---
**Algorithm 1:** Exponential weights (EXPWEIGHT)

---
1 **Initialize** $Y^0 \leftarrow 0$
2 **for** *t= 1, 2, . . .* **do**
3     set $X^t \leftarrow \Lambda(Y^{t-1})$ and get $C^t \leftarrow c(X^t, \omega^t)$        // route and measure costs
4     set $Y^t = Y^{t-1} - \gamma^t C^t$        // update path scores

---

Then, by applying the classical analysis of exponential weights methods [9, 46], we obtain the following equilibrium convergence guarantee:

**Theorem 1.** *Let $x^*$ be an equilibrium of $\Gamma$. If Algorithm 1 is run with variable learning rate $\gamma^t = 1/\sqrt{t}$, the time-averaged flow profile $\bar{X}^T = (1/T) \sum_{t=1}^{T} X^t$ enjoys the equilibrium convergence rate*

$$\mathbb{E}\big[F(\bar{X}^T) - F(x^*)\big] \leq \frac{N M_{\max}}{\sqrt{T}} \left[ \log\left(\frac{M_{\max} P}{M_{\text{tot}}}\right) + H^2(1 + \log T) \right] = \mathcal{O}\left(\frac{\log(PT)}{\sqrt{T}}\right). \quad (4)$$

This result confirms that EXPWEIGHT achieves a speed of convergence that is logarithmic in terms of $P$, and hence linear in the size $|\mathcal{G}| := |\mathcal{V}| + |\mathcal{E}|$ of the underlying network. Note that Theorem 1 is a direct application of standard results of EXPWEIGHT into the set-up of routing games (e.g., one can derive Theorem 1 from Corollary 2.14 and Theorem 4.1 of [46]). Since EXPWEIGHT is not the main focus of this work, we omit the details of the proof of Theorem 1; instead, we make two relevant remarks: First, the convergence rate obtained for Algorithm 1 concerns the empirical average flow, not the actual recommendation, a distinction which is important for practical applications. Second, there is a slight suboptimality in terms of the learning horizon $T$: the rate presented in Theorem 1 contains a $\log(T)$ term which can be shaved off by switching to the so-called "dual averaging" variant of the exponential weights template (or finetune the method's learning rate in terms of $T$ and subsequently use a doubling trick to obtain an anytime guarantee). For the details, we refer the reader to [38, 46, 50].

**3.2. Accelerated exponential weights in static environments.** We now turn our attention to the static regime, i.e., when there are no exogenous variations in the game's state ($\omega^t = \omega$ for all $t = 1, 2, \dots$). In this case, it is reasonable to expect that a faster convergence rate should be attainable: in particular, as we show below, the game's potential is Lipschitz smooth (see Proposition 2 below), so the optimal convergence speed in this case is the iconic $\mathcal{O}(1/T^2)$ rate of Nesterov [35]. In more detail, we have:

**Proposition 2.** *The BMW potential is Lipschitz smooth relative to the $L^1$ norm on $\mathcal{X}$ and has smoothness modulus $\beta = KL$, where $K$ is the length of the longest path in $\mathcal{P}$.*

Now, to maintain the graceful scaling guarantees of the EW template, our proposal to achieve an $\mathcal{O}(1/T^2)$ rate is an *accelerated exponential weights* algorithm that builds on ideas by Nesterov [35], Allen-Zhu and Orecchia [2] and Krichene et al. [24]. In pseudocode form, the method unfolds as follows:

---
**Algorithm 2:** Accelerated exponential weights (ACCELEWEIGHT)

---
**Input:** Smoothness parameter $\beta$
1 **Initialize** $Y^0 \leftarrow \mathbf{0}$, $X^0 = \mathbf{0}$, $\alpha^0 \leftarrow 0$ and $\gamma^0 \leftarrow \frac{1}{N M_{\max} \beta}$
2 **for** *t= 1, 2, . . .* **do**
3     set $Z^t \leftarrow \Lambda(Y^{t-1})$        // exploratory flow obtained from path scores
4     set $X^t \leftarrow \alpha^{t-1} X^{t-1} + (1 - \alpha^{t-1}) Z^t$        // average with previous state
5     set $\gamma^t \leftarrow \frac{1}{2}\left[2\gamma^{t-1} + \gamma^0 + \sqrt{4\gamma^{t-1}\gamma^0 + (\gamma^0)^2}\right]$        // update step-size
6     set $\alpha^t \leftarrow \gamma^{t-1}/\gamma^t$        // update moving weight
7     set $\bar{Z}^t \leftarrow \alpha^t X^t + (1 - \alpha^t) Z^t$ and get $C^t \leftarrow c(\bar{Z}^t, \omega^t)$        // route and measure costs
8     set $Y^t \leftarrow Y^{t-1} - (1 - \alpha^t)\gamma^t C^t$        // update path scores

---

Importantly, the step-size of Algorithm 2 is finetuned relative to the smoothness modulus of $F$, which is assumed known to the navigation interface. With this in mind, our main convergence guarantee for ACCELEWEIGHT is as follows:

**Theorem 2.** *Let $x^*$ be an equilibrium of $\Gamma$. If Algorithm 2 is run for $T$ epochs in the static regime, the traffic flow profile $X^T$ enjoys the equilibrium convergence rate:*

$$F(X^T) - F(x^*) \leq \frac{4\beta N^2 M_{\max}^2 \log(M_{\max} P/M_{\text{tot}})}{(T-1)^2} = \mathcal{O}\left(\frac{\log(P)}{T^2}\right). \tag{5}$$

Theorem 2 confirms that, in the static regime, ACCELEWEIGHT converges to equilibrium with an optimal $\mathcal{O}(\log(P)/T^2)$ convergence speed, as desired. The proof of Theorem 2 is based on techniques that are widespread in the analysis of accelerated methods, so we relegate it to Appendix C.

We only note here some limitations of the accelerated exponential weights method. First, the algorithm's convergence rate concerns a sequence of routing flows which is never recommended, a disparity which limits the algorithm's applicability. Second, the algorithm's step-size must be tuned with prior knowledge of the problem's smoothness modulus (which is not realistically available to the optimizer), so it is not adaptive in this regard. More importantly, ACCELEWEIGHT *fails to converge altogether in the stochastic regime*, so it does not achieve rate interpolation either. Particularly, ACCELEWEIGHT (and other Nesterov's acceleration methods) achieves convergence by building a consistent positive momentum towards a minimizer and then introducing a dissipative, "vanishing friction" term. In the stochastic regime, the fluctuations induced by the stochastic gradients fail to accumulate consistent momentum, so the acceleration provided is incoherent and hence the *non-convergence* of ACCELEWEIGHT.

**The per-iteration complexity of Algorithms 1 and 2.** In the discussions above, we chose to present simple implementations of EW and ACCELEWEIGHT (as Algorithms 1 and 2 respectively), in which every iteration runs in $\mathcal{O}(P)$ time. This is inefficient in large-scale networks where $P$ is typically exponentially large; however, owing to the underlying exponential weights template, both algorithms can be implemented efficiently in $\mathcal{O}(|\mathcal{E}|)$ time and space via a dynamic programming procedure known as "weight-pushing" [19, 47] under a mild assumption that the graphs defining the network are directed acyclic. The details of this efficient implementation lie beyond the scope of this work, so we do not present them here.

## 4   ADAWEIGHT: Adaptive learning in the presence of uncertainty

**4.1. Statement and discussion of results.** To summarize the situation so far, we have seen that EXPWEIGHT attains an $\mathcal{O}(\log(P)/\sqrt{T})$ rate, which is order-optimal in the stochastic case but suboptimal in static environments; by contrast, ACCELEWEIGHT attains an $\mathcal{O}(\log(P)/T^2)$ rate in static environment, but has no convergence guarantees in the presence of randomness and uncertainty. Consequently, neither of these algorithms meets our stated objective to concurrently achieve order-optimal guarantees in both the static and stochastic cases (and without requiring prior knowledge of the problem's smoothness modulus).

To resolve this gap, we propose below an *adaptive exponential weights* method – ADAWEIGHT for short – which achieves these objectives by mixing the acceleration template of ACCELEWEIGHT with the dual extrapolation method of Nesterov [37]. We present the pseudocode of ADAWEIGHT as Algorithm 3 below. The main novelty in the definition of the ADAWEIGHT algorithm is the introduction of two "extrapolation" sequences, $Z^t$ and $\tilde{Y}^t$, that venture outside the convex hull of the generated primal (flow) and dual (score) variables respectively. These leading states are subsequently averaged, and the method proceeds with an adaptive step-size rule. Particularly, ADAWEIGHT relies on three key components:

*a)* A dual extrapolation mechanism for generating the leading sequences in Lines 4 and 7; these sequences are central for anticipating the loss landscape of the problem.

*b)* An acceleration mechanism obtained from the $(\alpha^t)$-*weighted average* steps in Lines 5 and 8; in the analysis, $\alpha^t$ will grow as $t$, so almost all the weight will be attributed to the state closest to the current one.

*c)* An *adaptive sequence of learning rates* (cf. Line 10) in the spirit of Rakhlin and Sridharan [44], Kavis et al. [20] and Antonakopoulos et al. [3]. This choice is based on the ansatz that, if the algorithm encounters coherent gradient updates (which can only occur in static environments), it will eventually stabilize to a strictly positive value; otherwise, it will decrease to zero at a $\Theta(1/\sqrt{t})$ rate. This property is crucial to interpolate between the stochastic and static regimes.

---

**Algorithm 3:** adaptive exponential weights (ADAWEIGHT)

1   **Initialize** $\alpha^0 \leftarrow 0$, $Z^0 \leftarrow \mathbf{0}$, $\eta^1 \leftarrow 1$ and $Y^1 \leftarrow \mathbf{0}$

2   **for** $t = 1, 2, \ldots$ **do**

3      set $A^t \leftarrow \sum_{s=0}^{t-1} \alpha^s Z^s$                // set an anchor primal point

       // the test phase:

4      set $\tilde{Z}^t \leftarrow \Lambda(\eta^t Y^t)$                 // compute a pivot point

5      set $\tilde{X}^t \leftarrow \left(\alpha^t \tilde{Z}^t + A^t\right)/\sum_{s=0}^t \alpha^s$ and get $\tilde{C}^t \leftarrow c(\tilde{X}^t, \omega^t)$    // reweigh + query a test point

6      set $\tilde{Y}^t \leftarrow Y^t - \alpha^t \tilde{C}^t$             // exploratory score update

       // the recommendation phase:

7      set $Z^t \leftarrow \Lambda\left(\eta^t \tilde{Y}^t\right)$               // compute a pivot point

8      set $X^t \leftarrow \left(\alpha^t Z^t + A^t\right)/\sum_{s=0}^t \alpha^s$ and get $C^t \leftarrow c(X^t, \omega^t)$    // reweigh + route and measure costs

9      set $Y^{t+1} \leftarrow Y^t - \alpha^t C^t$             // update path scores

10      set $\eta^{t+1} \leftarrow 1/\sqrt{1 + \sum_{s=0}^t \left\|\alpha^s(C^s - \tilde{C}^s)\right\|_\infty^2}$       // update learning rate

---

The combination of the weighted average iterates and adaptive learning rate in ADAWEIGHT is shared by the UNIXGRAD algorithm proposed by Kavis et al. [20], which also provides rate interpolation in constrained problems. However, UNIXGRAD requires the problem's domain to have a finite Bregman diameter – and, albeit compact, the set of feasible flows $\mathcal{X}$ has an *infinite* diameter under the entropic regularizer that generates the EW template. Therefore, UNIXGRAD is not applicable to our routing games. This is the reason for switching gears to the "primal-dual" approach offered by the dual extrapolation template; this primal-dual interplay provides the missing link that allows ADAWEIGHT to simultaneously enjoy order-optimal convergence guarantees in both settings while maintaining the desired polynomial dependency on the problem's dimension.

In light of the above, our main convergence result for ADAWEIGHT is as follows:

**Theorem 3.** *Let $x^*$ be an equilibrium of $\Gamma$. If Algorithm 3 is run for $T$ epochs with $\alpha^t = t$ for any $t = 1, 2, \ldots$, the recommended flow profile $X^T$ enjoys the equilibrium convergence rate:*

$$\mathbb{E}\left[F(X^T) - F(x^*)\right] \le \frac{4\sqrt{2}\mathsf{A}KH}{\sqrt{T}} + \frac{16\beta\sqrt{NM_{\max}}\mathsf{A}^{3/2} + \mathsf{B}}{T^2} = \mathcal{O}\left(\frac{(\log P)^{3/2}}{\sqrt{T}}\right). \quad (6a)$$

*Moreover, in the static case, Algorithm 3 enjoys the sharper rate:*

$$F(X^T) - F(x^*) \le \frac{2\beta\mathsf{A}^{3/2} + \mathsf{B}}{T^2} = \mathcal{O}\left(\frac{(\log P)^{3/2}}{T^2}\right). \quad (6b)$$

*In the above expressions, $\mathsf{A}$ and $\mathsf{B}$ are positive constants given by $\mathsf{A} := NM_{\max}[2\log(PM_{\max}/M_{\mathrm{tot}}) + 13] = \mathcal{O}(\log P)$ and $\mathsf{B} := M_{\mathrm{tot}}\log(PM_{\max}/M_{\mathrm{tot}}) = \mathcal{O}(\log P)$.*

Theorem 3 confirms that ADAWEIGHT enjoys all of the desired features: (*i*) it achieves *simultaneously optimal guarantees* in both stochastic and static environments (i.e., $\mathcal{O}(1/\sqrt{T})$ and $\mathcal{O}(1/T^2)$ respectively); (*ii*) the derived rates maintain a *polynomial dependency* in terms of the network's combinatorial primitives; and (*iii*) it requires *no prior tuning* by the learner. Moreover, unlike EXPWEIGHT, the convergence of ADAWEIGHT corresponds to an actual traffic flow profile that is implemented in epoch $t$ and not the average flow.

**A smooth trade-off rate between static and stochastic environments.** In Inequality (6a) of Theorem 3, the convergence rate of ADAWEIGHT in the stochastic case is given in the worst-case scenario. Alternatively, one can also quantify the convergence speed of ADAWEIGHT according to the

level of randomness of the game. Particularly, define the random variables $\tilde{U}^t$, $U^t$ and $\mathtt{var}$ such that

$$\tilde{U}^t := \tilde{C}^t - \nabla F\left(\tilde{X}^t\right) \text{ and } U^t := C^t - \nabla F(X^t) \text{ and } \mathtt{var} := \max\left\{ \mathbb{E}\left[\|\tilde{U}^t\|_\infty^2\right], \mathbb{E}\left[\|U^t\|_\infty^2\right] \right\}. \quad (7)$$

Intuitively, $\tilde{U}^t$ and $U^t$ represent the noises of the observed costs $\tilde{C}^t$ and $C^t$, induced by $\tilde{X}^t$ and $X^t$, in comparison with the corresponding gradients of the BMW potential $F$; on the other hand, $\mathtt{var}$ quantifies the variances of these noises. With the same set-up as Theorem 3, the recommended flow $X^T$ of ADAWEIGHT enjoys the following equilibrium convergence rate: $\mathbb{E}\left[F\left(X^T\right) - F(x^*)\right] \leq \mathcal{O}\left((\log P)^{3/2}(\sqrt{\mathtt{var}}/\sqrt{T} + 1/T^2)\right)$. Interestingly, in special instances where the cost observations become more accurate over time, we can achieve a smooth trade-off in the convergence rate; e.g., if $\mathtt{var} = \mathcal{O}(1/T^\rho)$ for some suitable $\rho$, ADAWEIGHT 's rate of convergence carries a dependence of the order of $\mathcal{O}\left(T^{\max\{-1/2-\rho/2, -2\}}\right)$.

**Implementations of ADAWEIGHT.** As in the case of EXPWEIGHT and ACCELEWEIGHT, we note that ADAWEIGHT can also be implemented efficiently via the weight-pushing technique of Takimoto and Warmuth [47] (with linear space and time complexity in the size of the underlying graph). We leave the details of this efficient implementation for future works.

**Technical contributions of the proof of Theorem 3.** As analyzed above, ADAWEIGHT is *not* merely a "convex combination" of the two non-adaptive optimal algorithms (ACCELEWEIGHT and EXPWEIGHT). From this reason, the proof of Theorem 3 is technically involved. We analyze its main ideas in Section 4.2 and give a detailed proof in Appendix D.

**4.2. Sketch of proof of Theorem 3.** Let $\mathfrak{R}_T(x^*) := \sum_{t=1}^T t\langle\nabla F(X^t), Z^t - x^*\rangle$, the starting point of our proof is the following result:

$$\mathbb{E}\left[F(X^T) - F(x^*)\right] \leq 2\,\mathbb{E}[\mathfrak{R}_T(x^*)]/T^2 \text{ for any } T. \quad (8)$$

Intuitively, (8) shows that the convergence properties of ADAWEIGHT can be quantified via $\mathfrak{R}_T(x^*)$. Note that (8) is a standard result in working with dual-extrapolation and $\alpha$-weighted averaging techniques that also appears in several previous works [14, 21] (see Appendix D for a proof).

Recall that as ADAWEIGHT is run, $\nabla F(X^t)$ is *not* observable and only the cost $C^t$ is observed and used. Therefore, instead of $\mathfrak{R}_T(x^*)$, we now *focus on the term* $\mathfrak{R}_T^C(x^*) := \sum_{t=1}^T t\langle C^t, Z^t - x^*\rangle$. In the static case, $\mathfrak{R}_T^C(x^*)$ coincides with $\mathfrak{R}_T(x^*)$ and in the stochastic case, we can prove that $\mathfrak{R}_T^C(x^*)$ is an unbiased estimation thereof. Therefore, any upper-bound of $\mathfrak{R}_T^C(x^*)$ can be translated into an upper-bound of the left-hand-sides of (6a) and (6b) via (8). The key question becomes *"Which upper-bound of $\mathfrak{R}_T^C(x^*)$ can be derived to guarantee the convergence rates in (6a) and (6b)?"*

Recall that ADAWEIGHT (Algorithm 3) is built on the dual extrapolation template with two phases: the test phase and the recommendation phase. In the sequel, we aim to upper-bound $\mathfrak{R}_T^C(x^*)$ in terms of the "distance" between the pivot primal points in these phases (i.e., $\tilde{Z}^t$ and $Z^t$) and the difference between the costs measured in these phases (i.e., $\tilde{C}^t$ and $C^t$). Particularly, we have

$$\mathfrak{R}_T^C(x^*) \leq \sum_{t=1}^T \mathfrak{d}_{\text{dual}}(\eta^{t+1})t^2\left\|C^t - \tilde{C}^t\right\|_\infty^2 + \sum_{t=1}^T \mathfrak{d}_{\text{primal}}(\eta^{t+1})\|Z^t - \tilde{Z}^t\|_1^2, \quad (9)$$

where $\mathfrak{d}_{\text{dual}}(\eta^{t+1})$ and $\mathfrak{d}_{\text{primal}}(\eta^{t+1})$ are certain functions that also depend on other parameters of the game. Importantly, the terms in the right-hand-size of (9) are actually summable as follows:

(i) *In the static case*, it is bounded by the summation $\sum_{t=1}^T \mathfrak{d}_{\text{static}}(\eta^t)t^2\|C^t - \tilde{C}^t\|_\infty^2$ (due to the smoothness of the BMW potential $F$). Moreover, by our *special choice of adaptive sequence of learning rate $\eta^t$*, there exists $T_0 \ll T$ such that only the first $T_0$ components in this summation are positive and hence, this summation is actually of order $\mathcal{O}(1)$ (i.e., it does not depend on $T$).

(ii) *In the stochastic case*, beside the term derived in the static case, we also have an extra term in the upper-bound of (9) that has the form $\sqrt{\sum_{t=1}^T t^2\|[C^t - \tilde{C}^t] - [\nabla F(X^t) - \nabla F(\tilde{X}^t)]\|_\infty^2}$; intuitively, this quantifies the error of the observed costs with respect to the actual gradients of the potential. This term is of order $\mathcal{O}(T^{3/2})$.

Apply these result into (8), we obtain the convergence rates indicated in (6a) and (6b).

# 5 Numerical Experiments

**5.1. Experiments in the stochastic regime.** First, we conduct an experiment on a routing game with noisy observations (as described in Example 4) with the following setup: on a randomly generated network with 24 vertices and 276 edges, we assign to each edge a BPR cost function and choose $N = 6$ origin-destination pairs, each of which comes with a traffic demand; the traffic will be routed via the set of $P = 3366$ paths. Particularly, at each time $T$, after the algorithms decide their flow profiles, we perturb the induced costs on each edge $e$ by a noise $\omega_e^T$ that is drawn independently from the normal distribution $\mathcal{N}(0, 10)$. The algorithms only observe these stochastically perturbed costs and use them to update the next iterations.

For validation purposes, we run ExpWeight, AcceleWeight and AdaWeight in 5 instances (of noises' layout) and report, in Figure 1a, the averaged results across these instances. Particularly, we plot out the evolution of the following values: $\Delta_{\text{ExpWeight}}(T) \coloneqq F(\bar{X}^T) - F(x^*)$ where $\bar{X}^T$ is the time-averaged of outputs of Algorithm 1, $\Delta_{\text{AcceleWeight}}(T) \coloneqq F(X^T) - F(x^*)$ where $X^T$ is output by Algorithm 2 and $\Delta_{\text{AdaWeight}}(T) \coloneqq F(X^T) - F(x^*)$ where $X^T$ is output by Algorithm 3. Here, $x^*$ is chosen such that it induces the minimum $F$-value among all flow profiles computed by these algorithms after 15000 iterations; thus, $x^*$ represents the equilibrium flow.

Figure 1a shows that $\Delta_{\text{ExpWeight}}(T)$ and $\Delta_{\text{AdaWeight}}(T)$ tend to zero as $T$ increases; this confirms that ExpWeight and AdaWeight converge toward equilibrium. To this end, we observe that AcceleWeight fails to converge altogether. We also plot out $\sqrt{T} \cdot \Delta_{\text{AcceleWeight}}(T)$ and $\sqrt{T} \cdot \Delta_{\text{AdaWeight}}(T)$ (see Appendix E), and we observe that these terms approach horizontal lines as $T$ increases. This reaffirms the fact that the speed of convergence of ExpWeight and AdaWeight is $\mathcal{O}(1/\sqrt{T})$ in the stochastic regime which is consistent with our theoretical analyses.

**5.2. Experiments in static regime.** To illustrate the performance of the algorithms in the static case, we take advantage of a real data set collected and provided by [18] (with free license). This data set contains no personally identifiable information or offensive content. In this section, we present the experimental results on one instance in this data set: the SiouxFalls network (originated by [28]). Results corresponding to other network instances are presented in Appendix E. The SiouxFalls network has 24 vertices and 76 edges. We choose $N = 30$ origin-destination pairs and the total number of paths in use is $P = 3000$. We extract from the data set the BPR cost function corresponding to each edge and the traffic demand corresponding to each O/D pair. We run ExpWeight, AcceleWeight and AdaWeight in 16000 iterations and report the results thereof in Figure 1b.

Figure 1b confirms that in this static environment, all three algorithms converge towards equilibrium. Particularly, AcceleWeight and AdaWeight converges at an $\mathcal{O}(1/T^2)$ rate while ExpWeight fails to achieve this speed. These convergence rates are reaffirmed by observing that $T^2 \cdot \Delta_{\text{AcceleWeight}}(T)$ and $T^2 \cdot \Delta_{\text{AdaWeight}}(T)$ approach horizontal lines as $T$ increases (cf. Appendix E). We also run AcceleWeight with several initial step-size options and observe in Figure 1b that with badly-tuned parameters (i.e., with a wrongly estimation of the smoothness level of $F$), AcceleWeight might have a slow "warm-up" phase or might even diverge. By contrast, AdaWeight retains its appealing fast convergence properties throughout our experiments.

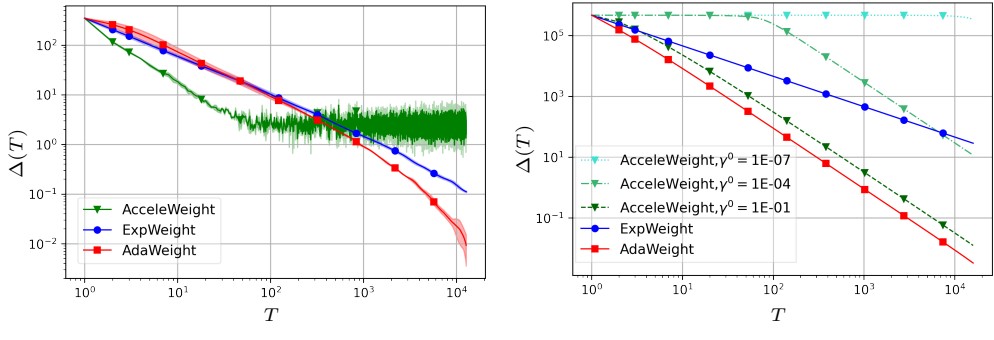

**(a)** Convergence in a stochastic environment     **(b)** Convergence in a static environment

**Figure 1:** The convergence speed of ExpWeight, AcceleWeight and AdaWeight in routing games.

## Acknowledgement

The authors are grateful for financial support by the French National Research Agency (ANR) in the framework of the "Investissements d'avenir" program (ANR-15-IDEX-02), the LabEx PERSYVAL (ANR-11-LABX-0025-01), MIAI@Grenoble Alpes (ANR-19-P3IA-0003), and the grant ALIAS (ANR-19-CE48-0018-01). The authors' research was also supported by the COST Action CA16228 "European Network for Game Theory" (GAMENET).

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
