# A  Auxiliary results and notions

The gradient of the function $F$ is important for our sequel analyses, we present its explicit computation in Appendix A.1. We also introduce several useful technical notions that will be used throughout the remainders of this work in Appendix A.2.

**A.1. Gradient of $F$.**  We notice that under Assumption 1, $c_e, e \in \mathcal{E}$ are continuous and non-decreasing functions. As a trivial consequence, $F$ is a convex function. Moreover, it is differentiable and when $\omega$ is generated randomly, for any $i \in \mathcal{N}$ and any path $p \in \mathcal{P}_i$, by recalling that $\mu_e(x) = \sum_{i \in \mathcal{N}} \sum_{\substack{p \in \mathcal{P}_i \\ p \ni e}} x_{i,p}$ and applying the dominated convergence theorem, we have:

$$
\frac{\partial F}{\partial x_{i,p}}(x) = \mathbb{E}\left[\frac{\partial F_\omega}{\partial x_{i,p}}(x)\right] = \mathbb{E}\left[\frac{\sum_{e \in \mathcal{E}} \int_0^{\mu_e(x)} c_e(\rho, \omega)\, d\rho}{\partial x_{i,p}}\right] = \mathbb{E}\left[\sum_{e \in \mathcal{E}} \frac{\partial \mu_e(x)}{\partial x_{i,p}} c_e(\mu_e(x), \omega)\right]
$$

$$
= \mathbb{E}\left[\sum_{e \in p} c_e(\mu_e(x), \omega)\right]
$$

$$
= C_p(x). \tag{A.1}
$$

Using the notation $c(x, \omega)$ to present the (random) vector of costs at flow $x$ and $\omega$, we can also rewrite that $\nabla F(x) = \mathbb{E}[c(x, \omega)]$ for any $x \in \mathcal{X}$.

## A.2.  Auxiliary notions.

**Entropy regularizer.**  In the remainder of this paper, we recurrently work with the function

$$
h(x) = \sum_{i \in \mathcal{N}} \sum_{p \in \mathcal{P}_i} x_{i,p} \log(x_{i,p}), \forall x \in \mathcal{X}. \tag{A.2}
$$

This function is often known as the *entropy regularizer*. It has several properties of interests:

- For any $x \in \mathcal{X}$, $h(x) \le \sum_{i \in \mathcal{N}} \sum_{p \in \mathcal{P}} x_{i,p} \log(M_i) \le M_{\mathrm{tot}} \log(M_{\max})$.
- $h$ is a $\frac{1}{\sigma}$-strongly convex function on $\mathcal{X}$ w.r.t the $\|\cdot\|_1$ norm where we denote $\sigma := M_{\max} N$.[1]
- The Fenchel conjugate of $h$—denoted by $h^*$—at an arbitrary $Y \in \mathbb{R}^P$ are:

$$
h^*(Y) := \max_{x \in \mathcal{X}} \langle x, Y \rangle - h(x) = \sum_{i \in \mathcal{N}} M_i \log\left(\sum_{q \in \mathcal{P}_i} \exp(Y_{i,q})\right),
$$

**KL divergence.**  Induced from the entropy regularizer $h$, we also define the Kullback–Leibler (KL) divergence between any flows $x, x' \in \mathcal{X}$ as follows:

$$
\mathrm{KL}(x \| x') := \sum_{i \in \mathcal{N}} \sum_{p \in \mathcal{P}_i} x_{i,p} \log\left(x_{i,p} / x'_{i,p}\right).
$$

In other words, $\mathrm{KL}(x \| x') = h(x) - h(x') - \langle \nabla h(x'), x - x' \rangle$. In the remainders of this work, we will also regularly use the following trivial equality regarding the KL-divergence:

$$
\mathrm{KL}(x \| x'') = \mathrm{KL}(x \| x') + \mathrm{KL}(x' \| x'') + \sum_{i \in \mathcal{N}} \sum_{p \in \mathcal{P}_i} \left(x'_{i,p} - x_{i,p}\right) \log\left(x''_{i,p} / x'_{i,p}\right). \tag{A.3}
$$

---

[1] Consider the functions $h_i((x_{i,p})_{p \in \mathcal{P}_i}) = \sum_{p \in \mathcal{P}_i} x_{i,p} \log(x_{i,p})$, it is trivial to see that $h_i(\cdot)$ is $\frac{1}{M_i}$-strongly convex on $\mathcal{X}_i = \left\{(x_{i,p})_{p \in \mathcal{P}_i} : \sum_{p \in \mathcal{P}_i} x_{i,p} = M_i\right\}$ w.r.t to the norm $\|\cdot\|_1$. Therefore, for any $x, x' \in \mathcal{X}$ and a sub-gradient $W \in \partial h(x')$, we have $h(x) = \sum_{i \in \mathcal{N}} h_i((x_{i,p})_{p \in \mathcal{P}_i}) \ge h(x') + \langle W, x - x' \rangle + \sum_{i \in \mathcal{N}} \frac{1}{2M_i}\left(\sum_{p \in \mathcal{P}} |x_{i,p} - x'_{i,p}|\right)^2 \ge h(x') + \langle W, x - x' \rangle + \frac{1}{2M_{\max} N}\|x - x'\|_1^2$ due to Cauchy-Schwarz inequality.

**Fenchel Coupling.** The *Fenchel coupling* between $x \in \mathcal{X}$ and $Y \in \mathbb{R}^P$ w.r.t the regularizer $h$ is defined as follows:

$$\mathcal{F}(x, Y) = h(x) + h^*(Y) - \langle x, Y \rangle. \tag{A.4}$$

**Smoothness alternative definition.** In previous sections, we use the definition that $g : \mathcal{K} \to \mathbb{R}$ is a $\beta$-smooth convex function over $\mathcal{K} \subset \mathbb{R}^d$ w.r.t the $\|\cdot\|$ norm if and only if $\|\nabla g(x) - \nabla g(x')\|_\infty \leq \beta \|x - x'\|, \forall x, x' \in \mathcal{K}$. This is also equivalent to the following conditions:

$$0 \leq g(x) - g(x') - \langle \nabla g(x'), x - x' \rangle \leq \frac{\beta}{2} \|x - x'\|^2, \forall x, x' \in \mathcal{K}. \tag{A.5}$$

# B  Proofs of Results in Section 2

## B.1. Proof of Proposition 1.

($\Rightarrow$)  Let $x^* \in \mathcal{X}$ be an equilibrium flow of the mean game $\Gamma$. For each $i \in \mathcal{N}$, let us define $C_i := \min_{\{q \in \mathcal{P}_i : x^*_{i,q} > 0\}} C_q(x^*)$. By definition of the equilibrium flow, for any $p \in \mathcal{P}_i$, we have

$$C_p(x^*) > C_i \text{ if } x^*_{i,p} = 0,$$
$$\text{and } C_p(x^*) = C_i \text{ if } x^*_{i,p} > 0.$$

Combining this with (A.1), for any arbitrary $x \in \mathcal{X}$, we have:

$$\langle \nabla F(x^*), x \rangle = \sum_{i \in \mathcal{N}} \sum_{p \in \mathcal{P}_i} \frac{\partial F}{\partial x_{i,p}}(x^*) x_{i,p} = \sum_{i \in \mathcal{N}} \sum_{p \in \mathcal{P}_i} C_p(x^*) x_{i,p} \geq \sum_{i \in \mathcal{N}} C_i \sum_{p \in \mathcal{P}_i} x_{i,p} = \sum_{i \in \mathcal{N}} C_i M_i,$$

Moreover, for $x^*$, we also obtain:

$$\langle \nabla F(x^*), x^* \rangle = \sum_{i \in \mathcal{N}} \sum_{p \in \mathcal{P}_i} \frac{\partial F}{\partial x_{i,p}}(x^*) x^*_{i,p} = \sum_{i \in \mathcal{N}} \sum_{\substack{p \in \mathcal{P}_i \\ x^*_{i,p} > 0}} C_p(x^*) x^*_{i,p} = \sum_{i \in \mathcal{N}} C_i \sum_{\substack{p \in \mathcal{P}_i \\ x^*_{i,p} > 0}} x^*_{i,p} = \sum_{i \in \mathcal{N}} C_i M_i.$$

In conclusion, we have $\langle \nabla F(x^*), x - x^* \rangle \geq 0$ for any $x \in \mathcal{X}$. Therefore, $x^*$ is a minimizer of $F$.

($\Leftarrow$)  Let $x^* \in \mathcal{X}$ be a minimizer of $F$. From the variational inequality corresponding to $F$, we have

$$\langle \nabla F(x^*), x - x^* \rangle \geq 0, \forall x \in \mathcal{X}$$
$$\Leftrightarrow \sum_{i \in \mathcal{N}} \sum_{p \in \mathcal{P}_i} C_p(x^*) x_{i,p} \geq \sum_{i \in \mathcal{N}} \sum_{p \in \mathcal{P}_i} C_p(x^*) x^*_{i,p}, \forall x \in \mathcal{X}. \tag{B.1}$$

We proceed by proof of contradiction. Assume that $x^*$ is *not* an equilibrium flow of $\Gamma$, i.e., there exist $j \in \mathcal{N}$ and $\hat{p}, \hat{q} \in \mathcal{P}_j$ such that

$$x^*_{j,\hat{p}} > 0 \text{ and } C_{\hat{p}}(x^*) > C_{\hat{q}}(x^*). \tag{B.2}$$

Consider the flow $x' \in \mathcal{X}$ that is defined as follows:

$$\begin{cases} x'_{j,p} = x^*_{j,p}, \forall p \in \mathcal{P}_j \backslash \{\hat{p}, \hat{q}\}, \\ x'_{j,\hat{p}} = x^*_{j,\hat{p}} - \delta, \\ x'_{j,\hat{q}} = x^*_{j,\hat{q}} + \delta, \\ x'_{i,p'} = x^*_{i,p'}, \forall i \in \mathcal{N} \backslash \{j\}, p' \in \mathcal{P}_i. \end{cases}$$

Here $\delta$ is taken such that $0 < \delta < x^*_{j,\hat{p}}$. Intuitively, $x'$ is the flow obtained by moving a $\delta$ amount of $j$-type traffic from the path $\hat{p}$ to the path $\hat{q}$ in the flow $x^*$.

By this definition, we have

$$\sum_{i \in \mathcal{N}} \sum_{p \in \mathcal{P}_i} C_p(x^*) x'_{i,p} - \sum_{i \in \mathcal{N}} \sum_{p \in \mathcal{P}_i} C_p(x^*) x^*_{i,p}$$

$$
\begin{aligned}
&= C_{\hat{p}}(x^*) \cdot (-\delta) + C_{\hat{q}}(x^*) \cdot (\delta) \\
&= \delta(C_{\hat{q}}(x^*) - C_{\hat{p}}(x^*)) \\
&< 0. \hspace{8cm} \text{(due to (B.2))}
\end{aligned}
$$

This contradicts (B.1); therefore, $x^*$ is a mean equilibrium flow of $\Gamma$. $\hspace{2cm}\square$

## C  Proofs of results in Section 3

In this section, we provide the proof of Proposition 2 showing the smooth-continuity of function $F$ in Appendix C.1, then in Appendix C.2, we prove Theorem 2 showing the convergence properties of the ACCELEWEIGHT algorithm .

### C.1.  Smoothness of $F$: Proof of Proposition 2.

*Proof.* From Assumption 1 and (A.1), for any $x, x' \in \mathcal{X}$, we have:

$$
\|\nabla F(x) - \nabla F(x')\|_\infty \leq \mathbb{E}[\|c(x,\omega) - c(x',\omega)\|_\infty] \leq \max_{\substack{i \in \mathcal{N} \\ p \in \mathcal{P}^i}} \mathbb{E}[\sum_{e \in p} L|\mu_e(x) - \mu_e(x')|]
$$

$$
\leq \mathbb{E}[L \sum_{e \in \mathcal{E}} \sum_{j \in \mathcal{N}} \sum_{\substack{q \in \mathcal{P}^j \\ q \ni e}} |x_q^j - x_q'^j|].
$$

For any $i \in \mathcal{N}$, denote the length of a path $p \in \mathcal{P}_i$ by $K_p$ (i.e., the number of edges containing in $p$), we have

$$
\|x - x'\|_1 = \sum_{j \in \mathcal{N}} \sum_{q \in \mathcal{P}^j} \sum_{e \in q} \frac{|x_q^j - x_q'^j|}{K_q} \geq \sum_{j \in \mathcal{N}} \sum_{q \in \mathcal{P}^j} \sum_{e \in q} \frac{|x_q^j - x_q'^j|}{K} = \frac{1}{K} \sum_{e \in \mathcal{E}} \sum_{j \in \mathcal{N}} \sum_{\substack{q \in \mathcal{P}^j \\ q \ni e}} |x_q^j - x_q'^j|.
$$

Combine the two arguments above, we have $\|\nabla F(x) - \nabla F(x')\|_\infty \leq \beta\|x - x'\|_1$ where $\beta := LK$. $\hspace{1cm}\square$

### C.2.  Convergence properties of the ACCELEWEIGHT algorithm: Proof of Theorem 2.
Theorem 2 considers the static regime where we consider a fixed game $\Gamma_\omega$ with a fixed $\omega \in \Omega$. In this setting, we also have $\nabla F(x) = c(x,\omega)$ for any flow $x \in \mathcal{X}$. Due to this reason, in this section, *we will use these notations interchangeably without further explanations*. In the following, we also use the entropy regularizer $h$ as defined in Appendix A.2 and let $X^t, Z^t, \gamma^{t-1}$ be defined as in Algorithm 2. We will prove Theorem 2 in the following 2 steps.

Step 1: *Define $\Delta^t := \gamma^{t-1}[F(X^t) - F(x^*)] + KL(x^*\|Z^t)$ and prove that $\Delta^t$ is a decreasing sequence as $t$ increases.* From Proposition 2 and the strong-convexity of $h$, we have

$$
\begin{aligned}
F(X^{t+1}) \leq & F(\bar{Z}^t) + \langle\nabla F(\bar{Z}^t), X^{t+1} - \bar{Z}^t\rangle + \frac{\beta}{2}\|X^{t+1} - \bar{Z}^t\|_1^2 \\
= & F(\bar{Z}^t) + \langle\nabla F(\bar{Z}^t), X^{t+1} - \bar{Z}^t\rangle + \frac{\beta}{2}(1-\alpha^t)^2\|Z^{t+1} - Z^t\|_1^2 \\
= & F(\bar{Z}^t) + \langle\nabla F(\bar{Z}^t), X^{t+1} - \bar{Z}^t\rangle + \beta\sigma(1-\alpha^t)^2 KL(Z^{t+1}\|Z^t). \hspace{1cm} \text{(C.1)}
\end{aligned}
$$

From (C.1) and the convexity of $F$, we have

$$
\begin{aligned}
& F(X^{t+1}) - F(x^*) - \alpha^t[F(X^t) - F(x^*)] \\
= & F(X^{t+1}) - [\alpha^t F(X^t) + (1-\alpha^t)F(x^*)] \\
\leq & F(X^{t+1}) - F(\bar{Z}^t) + \langle\nabla F(\bar{Z}^t), \bar{Z}^t - \alpha^t X^t - (1-\alpha^t)x^*\rangle \\
\leq & \langle\nabla F(\bar{Z}^t), X^{t+1} - \bar{Z}^t\rangle + \beta\sigma(1-\alpha^t)^2 KL(Z^{t+1}\|Z^t) + \langle\nabla F(\bar{Z}^t), \bar{Z}^t - \alpha^t X^t - (1-\alpha^t)x^*\rangle \\
= & (1-\alpha^t)\langle\nabla F(\bar{Z}^t), Z^{t+1} - x^*\rangle + \beta\sigma(1-\alpha^t)^2 KL(Z^{t+1}\|Z^t). \hspace{1cm} \text{(C.2)}
\end{aligned}
$$

Now, for the sake of brevity, let us denote $W^t = (1 - \alpha^t)\gamma^t c(\bar{Z}^t, \omega) = (1 - \alpha^t)\gamma^t \nabla F(\bar{Z}^t)$. From the update rules of Algorithm 2), we have:

$$\langle \nabla h(Z^t) - \nabla h(Z^{t+1}), Z^{t+1} - x^* \rangle$$

$$= \sum_{i \in \mathcal{N}} \log\left(\frac{\sum_{q \in \mathcal{P}_i} Z_{i,q}^t \exp(-W_q^t)}{Mi}\right) \sum_{p \in \mathcal{P}_i} \left(Z_{i,p}^{t+1} - x_{i,p}^*\right) + \sum_{i \in \mathcal{N}} \sum_{p \in \mathcal{P}} W_{i,p}^t \left(Z_{i,p}^{t+1} - x_{i,p}^*\right)$$

$$= 0 + \langle W^t, Z^{t+1} - x^* \rangle \qquad \text{(since } \sum_{p \in \mathcal{P}_i} Z_{i,p}^{t+1} = \sum_{p \in \mathcal{P}_i} x_{i,p}^* = Mi, \forall i \in \mathcal{N})$$

$$= (1 - \alpha^t)\gamma^t \langle \nabla F(\bar{Z}^t), Z^{t+1} - x^* \rangle. \tag{C.3}$$

Multiply two sides of (C.2) by $\gamma^t$ and combine with (C.3), we obtain that:

$$\gamma^t[F(X^{t+1}) - F(x^*)] - \gamma^t \alpha^t[F(X^t) - F(x^*)]$$

$$\leq \langle \nabla h(Z^t) - \nabla h(Z^{t+1}), Z^{t+1} - x^* \rangle + \gamma^t \sigma\beta(1 - \alpha^t)^2 \text{KL}(Z^{t+1} \| Z^t)$$

$$= \text{KL}(x^* \| Z^t) - \text{KL}(x^* \| Z^{t+1}) + \left(\gamma^t \sigma\beta(1 - \alpha^t)^2 - 1\right) \text{KL}(Z^{t+1} \| Z^t). \tag{C.4}$$

From the update rule of $\gamma^t$ in Line 5, the choice of $\gamma^0$ in Line 1 and the update rule of $\alpha^t$ in Line 6 of Algorithm 2, $\gamma^t \sigma\beta(1 - \alpha^t)^2 = \gamma^t \frac{1}{\gamma^0}\left(1 - \frac{\gamma^{t-1}}{\gamma^t}\right)^2 = \frac{(\gamma^{t-1} - \gamma^t)^2}{\gamma^t \gamma^0} = 1$. Therefore, the last term in (C.4) equals zero and we can rewrite (C.4) as:

$$\gamma^t[F(X^{t+1}) - F(x^*)] + \text{KL}(x^* \| Z^{t+1}) \leq \gamma^{t-1}[F(X^t) - F(x^*)] + \text{KL}(x^* \| Z^t).$$

This shows precisely that $\Delta^T \leq \Delta^{T-1} \leq \ldots \leq \Delta^1$ (by convention, we set $\gamma^{-1} = 0$). Particularly, we have:

$$\gamma^{T-1}[F(X^T) - F(x^*)] \leq \Delta^T \leq \Delta^{T-1} \leq \ldots \leq \Delta^1 = \text{KL}(x^* \| Z^1). \tag{C.5}$$

*Step 2: Upper-bounds of $F(X^T) - F(x^*)$:* To find such an upper-bound, we would like to use (C.5). To do this, we look for a lower-bound of $\gamma^{T-1}$ and an upper-bound of $\text{KL}(x^* \| X^1)$.

First, observe that $\sqrt{\gamma^{t-1}\sigma\beta} = \sqrt{\gamma^t \sigma\beta}\sqrt{1 - \frac{1}{\sqrt{\gamma^t \sigma\beta}}}$ and $\left(1 - \frac{1}{\sqrt{\sigma\beta\gamma^t}}\right)^{1/2} \leq 1 - \frac{1}{2\sqrt{\sigma\beta\gamma^t}}$, we obtain $\sqrt{\sigma\beta\gamma^t} \geq \sqrt{\sigma\beta\gamma^{t-1}} + 1/2$. Therefore,

$$\sqrt{\sigma\beta\gamma^{T-1}} \geq \sqrt{\sigma\beta\gamma^{T-2}} + \frac{1}{2} \geq \ldots \geq \sqrt{\sigma\beta\gamma^0} + \frac{T-1}{2} = \frac{T+1}{2} \geq \frac{T}{2}.$$

This implies that

$$\gamma^{T-1} > T^2/(4\sigma\beta) > 0. \tag{C.6}$$

Second, from the choice of $Y^0$ and $\alpha^0$ in Line 1 of Algorithm 2, we notice that $X_{i,p}^1 = Z_{i,p}^1 = M_i/P_i$ for any $i \in \mathcal{N}$ and $p \in \mathcal{P}_i$. Therefore, $\langle \nabla h(X^1), X^1 - x^* \rangle = 0$ and hence, $\text{KL}(x^* \| X^1) = h(x^*) - h(x^1)$. Moreover, recall that $h(x^*) \leq M_{\text{tot}} \log(M_{\text{max}})$, $-\min h \leq M_{\text{tot}} \log(P/M_{\text{tot}})$. From these results, we derive that

$$\text{KL}(x^* \| X^1) \leq M_{\text{tot}} \log(M_{\text{max}}) - \min_{x \in \mathcal{X}} h(x) \leq M_{\text{tot}} \log(PM_{\text{max}}/M_{\text{tot}}). \tag{C.7}$$

Combine (C.7), (C.6) into (C.5), then recall the notation $\sigma = M_{\text{max}}N$ (i.e., the strongly-convexity constant of $h$) and the fact that $M_{\text{tot}} = \sum_{i \in \mathcal{N}} Mi \leq NM_{\text{max}}$ we obtain that

$$F(X^T) - F(x^*) \leq \frac{4NM_{\text{tot}}M_{\text{max}}\beta \log\left(\frac{PM_{\text{max}}}{M_{\text{tot}}}\right)}{(T-1)^2} \leq \frac{4\beta(NM_{\text{max}})^2 \log\left(\frac{PM_{\text{max}}}{M_{\text{tot}}}\right)}{(T-1)^2}.$$

This concludes the proof. $\qquad\qquad\qquad\qquad\qquad\qquad\qquad\qquad\qquad\qquad\qquad\qquad\qquad\qquad\square$

# D Proofs of results in Section 4

In this section, we will provide the proof of Theorem 3 concerning the convergence properties of ADAWEIGHT (i.e., Algorithm 3). Before going into the details of these proofs, we prepare some useful results and lemmas.

**Remark.** In the remainder of this section, we let $\alpha^t = t, \forall t$ as chosen in Theorem 3.

**Lemma D.1.** *Let $(\rho^t)_{t \in \mathbb{N}}$ be a sequence of non-negative numbers, for any $T \in \mathbb{N}$, we have:*

$$\sqrt{\sum_{t=0}^{T} \rho^t} \leq \sum_{t=0}^{T} \frac{\rho^t}{\sqrt{\sum_{s=0}^{t} \rho^s}} \leq 2\sqrt{\sum_{t=0}^{T} \rho^t} \tag{D.1}$$

A proof of Lemma D.1 can be extracted from Lemma 5 of [32] (also re-stated as Lemma 2 of [21]).

**Lemma D.2.** *For any $X, X', Y, Y' \in \mathbb{R}^d$ and any norm $\|\cdot\|$, the following identity holds:*

$$\|X - X'\|_1 \|Y - Y'\|_\infty = \min_{\rho > 0} \left\{ \frac{\rho}{2} \|X - X'\|_1^2 + \frac{1}{2\rho} \|Y - Y'\|_\infty^2 \right\}. \tag{D.2}$$

*Proof.* For any $X, X', Y, Y' \in \mathbb{R}^d$, let us define the corresponding function $\psi : \mathbb{R} \to \mathbb{R}$ such that for any $\rho > 0$, $\psi(\rho) = \frac{\rho}{2} \|X - X'\|_1^2 + \frac{1}{2\rho} \|Y - Y'\|_\infty^2$ . Then, (D.2) comes from the fact that $\psi'(\rho^*) = 0$ and $\psi''(\rho^*) > 0$ where $\rho^* = \|X - X'\|_1 \|Y - Y'\|_\infty$. $\square$

Finally, recall the definition of $\Re_T(x)$ presented in Section 4.2, we prove Eq. (8):

*Proof.* Let us denote $\mathtt{R}^t := \sum_{s=1}^{t} s = \frac{t(t+1)}{2}$. By Line 8 of Algorithm 3, for any $t$, we have: $Z^t = \frac{\mathtt{R}^t}{t} X^t - \frac{\mathtt{R}^{t-1}}{t} X^{t-1}$. As a consequence,

$$
\begin{aligned}
\sum_{t=1}^{T} t \langle Z^t - x^*, \nabla F(X^t) \rangle &= \sum_{t=1}^{T} t \left\langle \frac{\mathtt{R}^t}{t} X^t - \frac{\mathtt{R}^{t-1}}{t} X^{t-1} - x^*, \nabla F(X^t) \right\rangle \\
&= \sum_{t=1}^{T} \left[ t \langle X^t - x^*, \nabla F(X^t) \rangle + \mathtt{R}^{t-1} \langle X^t - X^{t-1}, \nabla F(X^t) \rangle \right] \\
&\geq \sum_{t=1}^{T} t \left[ F(X^t) - F(x^*) \right] + \sum_{t=1}^{T} \mathtt{R}^{t-1} \left[ F(X^t) - F(X^{t-1}) \right] \\
&= T \left[ F(X^T) - F(x^*) \right] + \sum_{t=1}^{T-1} t \left[ F(X^T) - F(x^*) \right] \\
&= \sum_{t=1}^{T} t \left[ F(X^T) - F(x^*) \right]. \tag{D.3}
\end{aligned}
$$

Divide two sides of (D.3) by $\mathtt{R}^t > 0$ and notice that $\mathtt{R}^t \geq \frac{T^2}{2}$, we obtain that:

$$\frac{2}{T^2} \sum_{t=1}^{T} t \langle Z^t - x^*, \nabla F(X^t) \rangle \geq F(X^T) - F(x^*). \tag{D.4}$$

Taking the expectation of two sides of (D.4), we conclude the proof. $\square$

$\square$

**Proof of Theorem 3.** In the remainder of this section, we present the proof of Theorem 3. We will work with the *entropy regularizer* $h$ as defined in Appendix A.2. For the sake of brevity, we also define $\min h := \min_{x' \in \mathcal{X}} h(x)$ and denote the $\|\cdot\|_1$-diameter of $\mathcal{X}$ by $\varnothing := \max_{x,x' \in \mathcal{X}} \|x - x'\|_1$. First, we prove the following proposition (corresponding to Eq. (9) in the proof sketch):

**Proposition D.1.** *Run Algorithm 3, for any $x \in \mathcal{X}$, we have:*

$$\mathfrak{R}_T^C(x^*) \leq h(x^*) - \min h + \mathsf{A}_{\mathsf{dual}} \sum_{t=1}^{T} t^2 \eta^{t+1} \left\| C^t - \tilde{C}^t \right\|_\infty^2 - \sum_{t=1}^{T} \frac{1}{2\sigma \eta^{t+1}} \|Z^t - \tilde{Z}^t\|_1^2, \quad \text{(D.5)}$$

*where $\mathsf{A}_{\mathsf{dual}} := h(x^*) - \min h + \frac{\sigma^2 + 2\varnothing^2}{2\sigma}$.*

**Proof of Proposition D.1.** Consider an "intermediate" point $\mathcal{Q}^t := \Lambda\big(\eta^t Y^{t+1}\big)$, we focus on the terms $t\langle C^t, \mathcal{Q}^t - x^* \rangle$ and $t\langle C^t, Z^t - \mathcal{Q}^t \rangle$. These terms aggregate to $t\langle C^t, Z^t - x^* \rangle$ which defines $\mathfrak{R}_T^C(x^*)$.

First, we look for an upper bound of $t\langle C^t, \mathcal{Q}^t - x^* \rangle$; moreover, we desire that this upper-bound is summable with $t\langle C^t, Z^t - \mathcal{Q}^t \rangle$. From Line 9 of Algorithm 3, we have

$$
\begin{aligned}
t\langle C^t, \mathcal{Q}^t - x^* \rangle &= \frac{1}{\eta^t} \langle \eta^t Y^t - \eta^t Y^{t+1}, \mathcal{Q}^t - x^* \rangle \\
&= \frac{1}{\eta^t} \left[ -h(\mathcal{Q}^t) - h^*\big(\eta^t Y^{t+1}\big) + \langle x^*, \eta^t Y^{t+1} - \eta^t Y^t \rangle + \langle \mathcal{Q}^t, \eta^t Y^t \rangle \right] \\
&\leq \frac{1}{\eta^t} \left[ \mathcal{F}(x^*, \eta^t Y^t) - \mathcal{F}(x^*, \eta^t Y^{t+1}) - \mathrm{KL}\big(\mathcal{Q}^t \big\| \tilde{Z}^t\big) \right]. \quad \text{(D.6)}
\end{aligned}
$$

Here, the last inequality comes from the definition of Fenchel coupling and the fact that $\tilde{Z}^t = \Lambda(\eta^t Y^t)$. Now, apply the three-point inequality with Bregman divergence (Lemma 3.1 of [12]) to the KL-divergence,

$$
\begin{aligned}
\frac{1}{\eta^t} \mathrm{KL}\big(\mathcal{Q}^t \big\| \tilde{Z}^t\big) - \frac{1}{\eta^t} \mathrm{KL}\big(Z^t \big\| \tilde{Z}^t\big) - \frac{1}{\eta^t} \mathrm{KL}\big(\mathcal{Q}^t \big\| Z^t\big) &= \langle \nabla h(\tilde{Z}) - \nabla h(Z), Z^t - \mathcal{Q}^t \rangle \\
&\geq \mathrm{KL}\big(\mathcal{Q}^t \big\| \tilde{Z}^t\big) - \mathrm{KL}\big(Z^t \big\| \tilde{Z}^t\big) \\
&\geq t\big\langle \tilde{C}^t, Z^t - \mathcal{Q}^t \big\rangle.
\end{aligned}
$$

Combining this with (D.6), we derive that:

$$
\begin{aligned}
\mathfrak{R}_T^C(x^*) &= \sum_{t=1}^{T} t\langle C^t, \mathcal{Q}^t - x^* \rangle + t\langle C^t, Z^t - \mathcal{Q}^t \rangle \\
&\leq \underbrace{\sum_{t=1}^{T} \frac{1}{\eta^t} \left[ \mathcal{F}(x^*, \eta^t Y^t) - \mathcal{F}(x^*, \eta^t Y^{t+1}) \right]}_{:= \mathsf{A}_1} \\
&\quad + \underbrace{\left[ \sum_{t=1}^{T} t\big\langle C^t - \tilde{C}^t, Z^t - \mathcal{Q}^t \big\rangle - \sum_{t=1}^{T} \frac{1}{\eta^t} \mathrm{KL}(\mathcal{Q}^t \| Z^t) \right]}_{:= \mathsf{A}_2} - \underbrace{\sum_{t=1}^{T} \frac{1}{\eta^t} \mathrm{KL}\big(Z^t \big\| \tilde{Z}^t\big)}_{:= \mathsf{A}_3}.
\end{aligned}
$$
$$\text{(D.7)}$$

Now, we look for upper-bounds of the three terms in the right-hand-side of (D.7). First, we trivially have

$$\mathsf{A}_1 \leq \frac{1}{\eta^1} \mathcal{F}(x^*, \eta^1 Y^1) + \left( \frac{1}{\eta^{T+1}} - \frac{1}{\eta^1} \right)[h(x^*) - \min h] = \frac{1}{\eta^{T+1}}[h(x^*) - \min h]. \quad \text{(D.8)}$$

Second, for any $T$, from the Cauchy-Schwarz inequality and Lemma D.2,

$$\sum_{t=1}^{T} t\langle C^t - \tilde{C}^t, \mathcal{Q}^t - Z^t\rangle \leq \sum_{t=1}^{T}\left\|tC^t - t\tilde{C}^t\right\|_{\infty}\left\|\mathcal{Q}^t - Z^t\right\|_1$$

$$\leq \sum_{t=1}^{T}\left[\frac{t^2\sigma\eta^{t+1}}{2}\left\|C^t - \tilde{C}^t\right\|_{\infty}^2 + \frac{1}{2\sigma\eta^{t+1}}\left\|\mathcal{Q}^t - Z^t\right\|_1^2\right]. \qquad \text{(D.9)}$$

Combine this with the strong convexity of $h$, we obtain:

$$\mathsf{A}_2 \leq \frac{\sigma}{2}\sum_{t=1}^{T} t^2\eta^{t+1}\left\|C^t - \tilde{C}^t\right\|_{\infty}^2 + \frac{1}{2\sigma}\sum_{t=1}^{T}\left(\frac{1}{\eta^{t+1}} - \frac{1}{\eta^t}\right)\|\mathcal{Q}^t - \tilde{Z}^t\|_1^2$$

$$\leq \frac{\sigma}{2}\sum_{t=1}^{T} t^2\eta^{t+1}\left\|C^t - \tilde{C}^t\right\|_{\infty}^2 + \frac{\varnothing^2}{2\sigma}\left(\frac{1}{\eta^{T+1}} - 1\right). \qquad \text{(D.10)}$$

Third, use the strong convexity of $h$ again, we have

$$\mathsf{A}_3 \geq \sum_{t=1}^{T}\frac{1}{2\sigma\eta^t}\|Z^t - \tilde{Z}^t\|_1^2 = \sum_{t=1}^{T}\frac{1}{2\sigma\eta^{t+1}}\|Z^t - \tilde{Z}^t\|_1^2 - \frac{\varnothing^2}{2\sigma}\left(\frac{1}{\eta^{T+1}} - 1\right). \qquad \text{(D.11)}$$

Moreover, from the update rule of $\eta^{t+1}$ and Lemma D.1, we have:

$$\frac{1}{\eta^{T+1}} = \sqrt{1 + \sum_{t=1}^{T} t^2\|C^t - \tilde{C}^t\|_{\infty}^2} \leq \sum_{t=1}^{T}\frac{t^2\|C^t - \tilde{C}^t\|_{\infty}^2}{\sqrt{1 + \sum_{s=1}^{t} s^2\|C^s - \tilde{C}^s\|_{\infty}^2}} + 1$$

$$= \sum_{t=1}^{T} t^2\eta^{t+1}\|C^t - \tilde{C}^t\|_{\infty}^2 + 1. \qquad \text{(D.12)}$$

Combine (D.8), (D.10), (D.11) and (D.12) with (D.7), we obtain precisely (D.5). $\qquad \square$

Based on Proposition D.1, we now derive the convergence results of ADAWEIGHT. Particularly, we show that the second and third terms in (D.5) are actually summable.

**Proving** (6b) **– the convergence of ADAWEIGHT in the static case.** Consider the game $\Gamma_\omega$ where $\omega \in \Omega$ is fixed but unknown. In this setting, $\nabla F(x) = c(x, \omega)$ for any $x \in \mathcal{X}$; particularly, $\mathfrak{R}_T^C(x^*) = \mathfrak{R}_T(x^*)$, $\nabla F(\tilde{X}^t) = \tilde{C}^t$ and $\nabla F(X^t) = C^t$. Moreover, from Proposition 2 and the update rules of $\tilde{X}^t$ and $X^t$ in Lines 5 and 8 of Algorithm 3, $\left\|C^t - \tilde{C}^t\right\|_{\infty} = \|\nabla F(X^t) - \nabla F(\tilde{X}^t)\|_{\infty} \leq \beta\|X^t - \tilde{X}^t\|_1 = \beta\frac{t}{\sum_{s=1}^{t} s}\|Z^t - \tilde{Z}^t\|_1 \leq \frac{2\beta}{t}\|Z^t - \tilde{Z}^t\|_1$. Therefore, in the static regime, (D.5) becomes

$$\mathfrak{R}_T(x^*) \leq h(x^*) - \min h + \left(\mathsf{A}_{\mathsf{dual}} - \frac{1}{8\sigma(\eta^{t+1})^2\beta^2}\right)\sum_{t=1}^{T} t^2\eta^{t+1}\left\|C^t - \tilde{C}^t\right\|_{\infty}^2. \qquad \text{(D.13)}$$

Let us denote $T_0 = \max\{s \in \{1, \ldots, T\} : \eta^{s+1} \geq \left[2\beta\sqrt{2\mathsf{A}_{\mathsf{dual}}}\right]^{-1}\}$. Then, for any $t \geq T_0$, we have $\eta^{t+1}\left(\mathsf{A}_{\mathsf{dual}} - \frac{1}{8\sigma(\eta^{t+1})^2\beta^2}\right) < 0$. As a consequence, (D.5) can be re-written as follows:

$$\mathfrak{R}_T(x^*) \leq h(x^*) - \min h + \left(\mathsf{A}_{\mathsf{dual}} - \frac{1}{8\sigma(\eta^{t+1})^2\beta^2}\right)\sum_{t=1}^{T_0} t^2\eta^{t+1}\left\|C^t - \tilde{C}^t\right\|_{\infty}^2$$

$$\leq h(x^*) - \min h + \mathsf{A}_{\mathsf{dual}}\sum_{t=1}^{T_0}\frac{t^2\|C^t - \tilde{C}^t\|_{\infty}^2}{\sqrt{1 + \sum_{s=1}^{t} s^2\|C^s - \tilde{C}^s\|_{\infty}^2}}$$

$$\leq h(x^*) - \min h + \mathsf{A}_{\mathsf{dual}} 2\sqrt{1 + \sum_{t=1}^{T_0} t^2 \|C^t - \tilde{C}^t\|_\infty^2} \qquad \text{(by Lemma D.1)}$$

$$= h(x^*) - \min h + \mathsf{A}_{\mathsf{dual}} \frac{2}{\eta^{T_0+1}}$$

$$\leq h(x^*) - \min h + 4\sqrt{2}\beta(\mathsf{A}_{\mathsf{dual}})^{\frac{3}{2}}. \qquad (\text{D.14})$$

Now, it is trivial to verify that $h(x^*) \leq M_{\mathrm{tot}} \log(M_{\max})$ for any $x$ and $0 \leq -\min h \leq M_{\mathrm{tot}} \log(P/M_{\mathrm{tot}})$; moreover, $M_{\mathrm{tot}} \leq NM_{\max}$, $\sigma := NM_{\max}$ and $\o \leq 2M_{\mathrm{tot}}$. Plug these results into (D.14), we obtain:

$$\mathfrak{R}_T(x) \leq M_{\mathrm{tot}} \log(PM_{\max}/M_{\mathrm{tot}}) + 2\beta \left[ NM_{\max}\left( 2\log\left(\frac{PM_{\max}}{M_{\mathrm{tot}}}\right) + 9 \right) \right]^{\frac{3}{2}}. \qquad (\text{D.15})$$

Finally, for the sake of conciseness in presentation of Theorem 3, we also note that $NM_{\max}\left( 2\log\left(\frac{PM_{\max}}{M_{\mathrm{tot}}}\right) + 9 \right) \leq \mathsf{A} := NM_{\max}\left( 2\log\left(\frac{PM_{\max}}{M_{\mathrm{tot}}}\right) + 13 \right)$ (this constant $\mathsf{A}$ will appear in the proof of (6a) concerning the stochastic case). Combine (D.15) with (8), we obtain (6b) and conclude the proof for the static case.

**Proving (6a) – the convergence of ADAWEIGHT in the stochastic case.** In this setting, we recall the notation $U^t$, $\tilde{U}^t$ and var from (7). From these definitions, we also deduce that $\mathbb{E}\left[\tilde{U}^t\middle|\mathcal{H}^{t-1}\right] = \mathbb{E}\left[U^t\middle|\mathcal{H}^{t-1}\right] = \mathbf{0}$ where $\mathcal{H}^{t-1} = \left\{ X^{t-1}, \tilde{X}^{t-1}, \omega^{t-1}, \ldots, X^1, \tilde{X}^1, \omega^1 \right\}$ is the filtration up to time epoch $t-1$.

Unlike the static regime where we can directly utilize the results related to $\nabla F$ to analyze ADAWEIGHT, in the stochastic case, there is a gap between the difference of the actual gradient of the BMW potential $\nabla F(X^t) - \nabla F(\tilde{X}^t)$ and the term $C^t - \tilde{C}^t$ involving in (D.5). We define the following terms in order to analyze this gap:

$$\mathfrak{D}^t = \min\left\{ \|\nabla F(X^t) - \nabla F(\tilde{X}^t)\|_\infty^2, \|C^t - \tilde{C}^t\|_\infty^2 \right\} \text{ and } \xi^t = \left[ C^t - \tilde{C}^t \right] - \left[ \nabla F(X^t) - \nabla F(\tilde{X}^t) \right].$$

We aim to construct upper-bounds of the last two terms in the right-hand-side of (D.5) in terms of $\mathfrak{D}^t$ and $\xi^t$. Particularly, from (D.5) (i.e., Proposition D.1), we can prove the following proposition:

**Proposition D.2.** *Run Algorithm 3 and define $\tilde{\eta}^t := 1/\sqrt{1 + 2\sum_{s=1}^{t-1} s^2 \mathfrak{D}^s}$, we have*

$$\mathfrak{R}_T^C(x^*) \leq h(x^*) - \min h + \mathsf{A}_{\mathsf{stoch}} 2\sqrt{2}\left( \sum_{t=1}^T t^2 \|\xi_t\|_\infty^2 \right)^{1/2} + \sum_{t=1}^T \eth(\tilde{\eta}^{t+1}) t^2 \mathfrak{D}^t, \qquad (\text{D.16})$$

*where $\mathsf{A}_{\mathsf{stoch}} := h(x^*) - \min h + \frac{\sigma^2 + 3\o^2}{2\sigma}$ and $\eth(\tilde{\eta}^{t+1}) := 4\mathsf{A}_{\mathsf{stoch}}\tilde{\eta}^{t+1} - \frac{1}{8\beta^2\sigma\tilde{\eta}^{t+1}}$.*

*Proof of Proposition D.2.* In order to prove (D.16), we first analyze the terms $\sum_{t=1}^T t^2\eta^{t+1}\|C^t - \tilde{C}^t\|_\infty^2$ and $\sum_{t=1}^T \frac{1}{2\sigma\eta^{t+1}}\|Z^t - \tilde{Z}^t\|_1^2$ that appear in (D.5). First, by definition of $\mathfrak{D}^t$ and $\xi^t$,

$$\|C^t - \tilde{C}^t\|_\infty^2 = \mathfrak{D}^t + \left[ \|C^t - \tilde{C}^t\|_\infty^2 - \min\left\{ \|\nabla F(X^t) - \nabla F(\tilde{X}^t)\|_\infty^2, \|C^t - \tilde{C}^t\|_\infty^2 \right\} \right]$$

$$\leq \mathfrak{D}^t + \max\{0, \|C^t - \tilde{C}^t\|_\infty^2 - \|\nabla F(X^t) - \nabla F(\tilde{X}^t)\|_\infty^2\}$$

$$\leq 2\mathfrak{D}^t + 2\|\xi^t\|_\infty^2 \qquad (\text{D.17})$$

Applying Lemma D.1, we obtain that

$$\sum_{t=1}^T t^2\eta^{t+1}\|C^t - \tilde{C}^t\|_\infty^2 = \sum_{t=1}^T \frac{t^2\|C^t - \tilde{C}^t\|_\infty^2}{\sqrt{1 + \sum_{s=1}^t s\|C^s - \tilde{C}^s\|_\infty^2}}$$

$$\leq 2 \sqrt{1 + \sum_{t=1}^{T} t^2 \|C^t - \tilde{C}^t\|_{\infty}^2}$$

$$\leq 2 \sqrt{1 + 2 \sum_{t=1}^{T} t^2 \mathfrak{D}^t + 2 \sum_{t=1}^{T} t^2 \|\xi^t\|_{\infty}^2}$$

$$\leq \sqrt{1 + 2 \sum_{t=1}^{T} t^2 \mathfrak{D}^t} + \sqrt{2 \sum_{t=1}^{T} t^2 \|\xi^t\|_{\infty}^2}$$

$$\leq 4 \sum_{t=1}^{T} \frac{t^2 \mathfrak{D}^t}{\sqrt{1 + 2 \sum_{i=1}^{t} \alpha_i^2 \mathfrak{D}^t}} + \sqrt{2 \sum_{t=1}^{T} t^2 \|\xi^t\|_{\infty}^2}. \tag{D.18}$$

On the other hand, from the definition of $\tilde{\eta}^t$, we have $\frac{1}{\tilde{\eta}^t} \leq \frac{1}{\eta^t}, \forall t$ and hence,

$$\sum_{t=1}^{T} \frac{1}{\tilde{\eta}^{t+1}} \left\| \tilde{Z}^t - Z^t \right\|_1^2 \leq \sum_{t=1}^{T} \left( \frac{1}{\eta^{t+1}} - \frac{1}{\eta^t} \right) \left\| Z^t - \tilde{Z}^t \right\|_1^2 + \frac{1}{\eta^t} \left\| Z^t - \tilde{Z}^t \right\|_1^2$$

$$\leq \left( \frac{1}{\eta^{T+1}} - 1 \right) \varnothing^2 + \sum_{t=1}^{T} \frac{1}{\eta^t} \left\| Z^t - \tilde{Z}^t \right\|_1^2.$$

Combine this with the fact that $Z^t - \tilde{Z}^t = \frac{(t+1)}{2} \left( X^t - \tilde{X}^t \right)$ (from Lines 5 and 8 of Algorithm 3), we have

$$-\sum_{t=1}^{T} \frac{1}{2\sigma \eta^{t+1}} \|Z^t - \tilde{Z}^t\|_1^2 \leq -\frac{1}{2\sigma} \sum_{t=1}^{T} \frac{1}{\eta^t} \|Z^t - \tilde{Z}^t\|_1^2 \qquad \text{(since } \eta^{t+1} \leq \eta^t)$$

$$\leq -\frac{1}{2\sigma} \sum_{t=1}^{T} \frac{1}{\tilde{\eta}^{t+1}} \left\| Z^t - \tilde{Z}^t \right\|_1^2 + \frac{\varnothing^2}{2\sigma} \left( \frac{1}{\eta^{T+1}} - 1 \right)$$

$$\leq -\frac{1}{2\sigma} \sum_{t=1}^{T} \frac{(t+1)}{2\tilde{\eta}^{t+1}} \left\| X^t - \tilde{X}^t \right\|_1^2 + \frac{\varnothing^2}{2\sigma} \left( \frac{1}{\eta^{T+1}} - 1 \right)$$

$$\leq -\frac{1}{2\sigma} \sum_{t=1}^{T} \frac{1}{\tilde{\eta}^{t+1}} \frac{(t+1)^2 \left\| \nabla F(X^t) - \nabla F(\tilde{X}^t) \right\|_{\infty}^2}{4\beta^2} + \frac{\varnothing^2}{2\sigma} \left( \frac{1}{\eta^{T+1}} - 1 \right)$$

$$\leq -\frac{1}{8\beta^2 \sigma} \sum_{t=1}^{T} \frac{t^2 \mathfrak{D}^t}{\tilde{\eta}^{t+1}} + \frac{\varnothing^2}{2\sigma} \left( \frac{1}{\eta^{T+1}} - 1 \right). \tag{D.19}$$

Apply (D.18) and (D.19) into (D.5), we obtain (D.16) and finish the proof of Proposition D.2. $\quad\square$

Denote $T_0 := \max \left\{ 1 \leq t \leq T : \tilde{\eta}^{t+1} \geq \left[ 32\beta^2 \sigma \mathsf{A}_{\mathsf{stoch}} \right]^{-1/2} \right\}$. Then for any $t \geq T_0$, we have $\eth(\tilde{\eta}^{t+1}) := 4\mathsf{A}_{\mathsf{stoch}} \tilde{\eta}^{t+1} - \frac{1}{8\beta^2 \sigma \tilde{\eta}^{t+1}} < 0$. In other words, in $\sum_{t=1}^{T} \eth(\tilde{\eta}^{t+1}) t^2 \mathfrak{D}^t$, only the first $T_0$ components are positive. Therefore, using the same trick (with $T_0$) leading to (D.14), we have:

$$\sum_{t=1}^{T} \eth(\tilde{\eta}^{t+1}) t^2 \mathfrak{D}^t \leq \sum_{t=1}^{T_0} 4\mathsf{A}_{\mathsf{stoch}} \tilde{\eta}^{t+1} t^2 \mathfrak{D}^t \leq \frac{8}{\tilde{\eta}^{T_0}} \mathsf{A}_{\mathsf{stoch}} \leq 32\beta\sqrt{2\sigma}(\mathsf{A}_{\mathsf{stoch}})^{3/2}. \tag{D.20}$$

Combine (D.20), (D.16) and the fact that $\mathbb{E}\left[ \|\xi_t\|_{\infty}^2 \right] \leq 2\mathbb{E}\left[ \|U^t\|_{\infty}^2 \right] + 2\mathbb{E}\left[ \|\tilde{U}^t\|_{\infty}^2 \right] \leq 4\mathsf{var}$, we have

$$\mathbb{E}\left[ \mathfrak{R}_T^C(x^*) \right] \leq h(x^*) - \min h + \mathsf{A}_{\mathsf{stoch}} 4\sqrt{2\mathsf{var}} \left( \sum_{t=1}^{T} t^2 \right)^{1/2} + 32\beta\sqrt{2\sigma}(\mathsf{A}_{\mathsf{stoch}})^{3/2}. \tag{D.21}$$

Finally, in order to apply (8), we need to make the connection between $\mathbb{E}[\mathfrak{R}_T(x^*)]$ and $\mathbb{E}\left[\mathfrak{R}_T^C(x^*)\right]$. Particularly, we have:

$$
\begin{aligned}
\mathbb{E}[\mathfrak{R}_T(x^*)] &= \mathbb{E}\left[\mathfrak{R}_T^C(x^*)\right] - \mathbb{E}\left[\sum_{t=1}^T t\langle C^t - \nabla F(X^t), Z^t - x^*\rangle\right] \\
&= \mathbb{E}\left[\mathfrak{R}_T^C(x^*)\right] - \mathbb{E}\left[\sum_{t=1}^T t\langle U^t, Z^t - x^*\rangle\right] \\
&= \mathbb{E}\left[\mathfrak{R}_T^C(x^*)\right].
\end{aligned}
\tag{D.22}
$$

Here, the last equality comes from the fact that $\mathbb{E}\left[\sum_{t=1}^T t\langle U^t, Z^t - x^*\rangle\right] = \mathbb{E}\left[\sum_{t=1}^T t\langle\mathbb{E}[U^t|\mathcal{H}^{t-1}], Z^t - x^*\rangle\right] = 0$ (by the law of total expectation). Combine (D.22) and (D.21), then apply (8), we have:

$$
\mathbb{E}\left[F(X^T) - F(x^*)\right] \leq \frac{1}{T^2}\left(h(x^*) - \min h + \mathsf{A}_{\mathsf{stoch}}4\sqrt{2\mathsf{var}}T^{3/2} + 32\beta\sqrt{2\sigma}(\mathsf{A}_{\mathsf{stoch}})^{3/2}\right).
\tag{D.23}
$$

Finally, recall that $h(x^*) \leq M_{\mathrm{tot}}\log(M_{\max})$, $-\min h \leq M_{\mathrm{tot}}\log(P/M_{\mathrm{tot}})$, $\sigma := NM_{\max}$, $\varnothing \leq 2M_{\mathrm{tot}}$ and note that $\mathsf{var} \leq 4K^2H^2$, we have $\mathsf{A}_{\mathsf{stoch}} \leq \frac{1}{2}NM_{\max}\left[2\log\left(\frac{PM_{\max}}{M_{\mathrm{tot}}}\right) + 13\right] := \frac{1}{2}\mathsf{A}$. Plug these results into (D.23), we obtain (6a).

## E  Supplementary numerical experiments

The code of our numerical experiments are available at `https://github.com/dongquan-vu/Adaptive_Routing_ExpWeight_Neurips2021`.

For illustrating the convergence properties of the EXPWEIGHT, ACCELEWEIGHT and ADAWEIGHT algorithms, we present below more experimental results, in addition to the ones discussed in Section 5. Particularly, in Appendix E.1, we report the experiments conducted in a stochastic setting and in Appendix E.2, we report the experiments done in a static environment.

**E.1. Performance in the stochastic regime.**   In this section, we work with the stochastic regime as described in Section 5, i.e., in a game with stochastically perturbed observations. Theorem 3 guarantees that ADAWEIGHT converges at an $\mathcal{O}(1/\sqrt{T})$ rate. Since Theorem 3 is our core result, we re-justify it by plotting out the evolution of the term $\sqrt{T}\cdot\Delta_{\mathrm{ADAWEIGHT}}(T)$ (defined in Section 5). We also plot out $\sqrt{T}\cdot\Delta_{\mathrm{EXPWEIGHT}}(T)$, $\sqrt{T}\cdot\Delta_{\mathrm{ACCELEWEIGHT}}(T)$ for the sake of comparison and report all of these results in Figure 2.

Figure 2 shows that $\sqrt{T}\cdot\Delta_{\mathrm{ADAWEIGHT}}(T)$ approaches a horizontal line as $T$ increases. This re-confirms that the convergence rate of ADAWEIGHT is $\mathcal{O}(1/\sqrt{T})$. On the other hand, the divergence of $\sqrt{T}\cdot\Delta_{\mathrm{ACCELEWEIGHT}}(T)$ confirms once again that ACCELEWEIGHT does not converge in this setting.

In the experiment reported in Figure 1 and Figure 2, the induced costs on each edge $e$ at each time epoch $t$ is perturbed by a noise $\omega_e^T$ that is drawn independently from the normal distribution $\mathcal{N}(0, 10)$. Now, to study the effect of noises on the performance of the algorithms, we also re-run this experiment in a "noisier" setting: the noise $\omega_e^T$ is generated from $\mathcal{N}(0, 50)$ for any $T$ and any $e$. We report the results in Figure 3. At a high-level, we see that ADAWEIGHT is more stable in the presence of noise with high-variance than EXPWEIGHT (or ACCELEWEIGHT).

**E.2. Performance in the static regime.**   In this section, we work with the static setting. SiouxFalls is a famous network that is often used as a benchmark in the field and we reported, in Section 5, the performance of the EXPWEIGHT, ACCELEWEIGHT and ADAWEIGHT algorithms on this network with data extracted from the real data set [18]. To support further this numerical experiments and to re-justify the theoretical convergence rates guaranteed in Theorem 1 and Theorem 3, we plot out

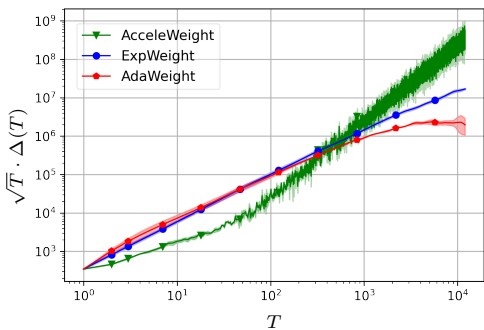

**Figure 2:** The order of the convergence rates in a stochastic environment.

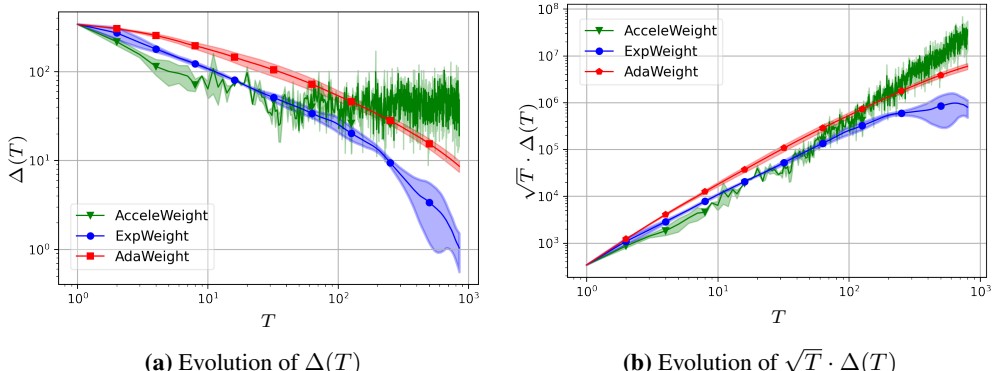

(a) Evolution of $\Delta(T)$

(b) Evolution of $\sqrt{T} \cdot \Delta(T)$

**Figure 3:** The convergence speed of EXPWEIGHT, ACCELEWEIGHT and ADAWEIGHT in the stochastic regime *with high-level of noises*.

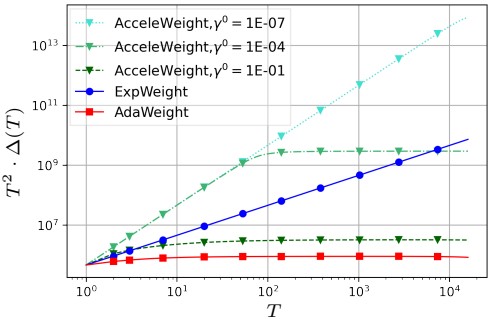

**Figure 4:** The order of the convergence rates in the static regime.

$T^2 \cdot \Delta_{\text{EXPWEIGHT}}(T)$, $T^2 \cdot \Delta_{\text{ACCELEWEIGHT}}(T)$ and $T^2 \cdot \Delta_{\text{ADAWEIGHT}}(T)$. These results are reported in Figure 4. It shows that $T^2 \cdot \Delta_{\text{ACCELEWEIGHT}}(T)$ and $T^2 \cdot \Delta_{\text{ADAWEIGHT}}(T)$ approach horizontal lines as $T$ increases; this confirms that ACCELEWEIGHT and ADAWEIGHT converge at an $\mathcal{O}(1/T^2)$ rate which are in consistent with Theorem 2 and Theorem 3. On the other hand, $T^2 \cdot \Delta_{\text{EXPWEIGHT}}(T)$ increases (almost) linearly; this demonstrates that the EXPWEIGHT algorithm fails to achieve an $\mathcal{O}(1/T^2)$-convergence rate in this setting. The slow-convergence of ACCELEWEIGHT with bad-tuned step-sizes (e.g., when $\gamma^0 = 1e-07$) is also re-confirmed in Figure 4

Finally, we show in Figure 5 the experiments results with respected to several other networks in the Transportation Networks data set (provided by [18]) that are larger than SiouxFalls. We observe that, at a high level, the performance of EXPWEIGHT, ACCELEWEIGHT and ADAWEIGHT in these networks is similar to that in the SiouxFalls network.

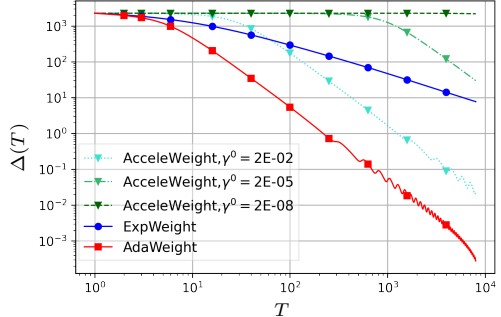

(a) Eastern-Massachusetts network: 74 vertices, 258 edges; we choose $N = 50$ O/D pairs, each of which is assigned with 10 shortest paths; the total paths in used is $P = 500$.

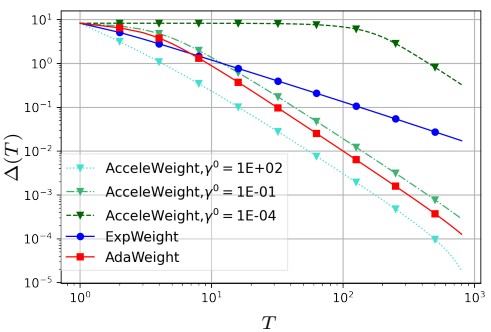

(b) Austin network: 7388 vertices, 18961 edges; we choose $N = 5$ O/D pairs which is assigned with 10 shortest paths; the total paths in used is $P = 50$.

**Figure 5:** The convergence speed of EXPWEIGHT, ACCELEWEIGHT and ADAWEIGHT algorithms in the static regime in Eastern-Massachusetts and Austin networks.