# OpenReview forum: "Fast Routing under Uncertainty: Adaptive Learning in Congestion Games via Exponential Weights"
_NeurIPS.cc/2021/Conference — NeurIPS 2021 Poster_

### Official Review · Reviewer_U9xb · 2021-06-25

**Rating:** 6
**Confidence:** 3

**Summary:**

This paper studies the equilibrium computation of congestion games and proposes a new algorithm, named ADAWEIGHT, that achieves $O(1/\sqrt{T})$ convergence in the stochastic regime and $O(1/T^2)$ in the static regime. The algorithm has several desirable properties:

1. it attains the best rates in the stochastic and static regimes,
2. the rates depend on $\log P$, not on $P$,
3. it is parameter-agnostic (no knowledge of parameters is needed),
4. it enjoys any-time guarantee (i.e., last-iterate convergence).

The results are obtained using some tools: primal and dual extrapolation, weighted-average analysis, and an adaptive learning rate. Experiments confirm the fast convergence of ADAWEIGHT in both regimes, while the existing ACCELEWEIGHT fails to converge in the stochastic regime.

--- After rebuttal ---

I appreciate the authors' kind response. I think the paper provides solid technical results, while the contribution to the ML community is somewhat marginal. Taking those into account, I keep my score of 6. I think the paper would be strengthened if the authors could work more on the smooth trade-off between the stochastic and static regimes.

**Limitations And Societal Impact:**

The limitations are described. The paper seems to have no negative societal impacts.

**Main Review:**

The paper is clearly written, and the proposed algorithm is well-positioned in the literature. The proposed algorithm has considerable advantages, as summarized above. On the other hand, I have several concerns and questions as listed below:


1. It is shown only experimentally that ACCELEWEIGHT fails to converge in the stochastic regime. Why is it theoretically difficult to guarantee convergence of accelerated methods in the stochastic regime? In my opinion, clarifying this will help improve the significance of the theoretical result.

2. In the experiments, UNIXGRAD, whose convergence rates are the same as ADAWEIGHT regarding $T$ in both regimes, is not presented. What if it is compared with ADAWEIGHT?

3. In (Wang & Abernethy NeurIPS 2018, "Acceleration through Optimistic No-Regret Dynamics"), a similar analysis of $\alpha$-weighted regret is presented. I would like the authors to explain whether there is a connection between the $(\alpha^t)$-weighted average technique and that of Wang & Abernethy.

4. In my understanding, weight pushing is applicable only to DAGs. This should be mentioned in the last paragraph in Section 3.2 (I would appreciate it if the authors could tell me if I am wrong).

5. Is it possible to extend the proposed algorithm to obtain a parameterized guarantee such that, for $\rho\in[1/2, 2]$ representing some degree of randomness, $O(1/T^\rho)$ is achieved?

6. In practice, the distribution, $\mathbb{P}$, may also change over time. Could you say something in such cases?

Minor comments:

- It should be noted that if potential functions are strongly convex, $O(\exp(-T))$ convergence is possible in the static regime, e.g., (Nakamura, Sakaue, Yasuda, AAAI 2020, "Practical Frank–Wolfe Method with Decision Diagrams for Computing Wardrop Equilibrium of Combinatorial Congestion Games").

- Big O notation sometimes hides the $\log T$ factor of EXPWEIGHT (e.g., in the first sentence in Section 4.1). It is better to use a different notation, e.g., $\mathcal{O}^*$, for clarity.

**Time Spent Reviewing:**

3

---

> ### Author Response · Authors · 2021-08-10
> **Reply to Reviewer U9xb**
>
>
> Dear reviewer,
>
> Thank you for your time and comments. We provide a point-by-point reply to your questions below:
>
> 1. On the non-convergence of AcceleWeight in stochastic environments. Nesterov’s acceleration method achieves convergence by building a consistent positive momentum towards a minimizer and then introducing a dissipative, “vanishing friction” term. In the stochastic case, the $\sqrt{t}$ fluctuations induced by the stochastic gradients fail to accumulate consistent momentum towards a minimizer, so the acceleration provided is incoherent with the problem in hand. As a result, it is well-known that accelerated methods fail to converge---as do all methods with a non-vanishing step-size---in the presence of persistent gradient noise.
>
> 2. On simulations for UnixGrad. Because the simplex has an infinite entropic diameter, UnixGrad fails to converge altogether. We observed this breakdown empirically in our numerical experiments, but we did not report it because it would be unfair to UnixGrad (which was never designed to solve this type of problems anyway). That being said, if the reviewer feels that this would lead to a more complete assessment, we would be happy to include UnixGrad in our experiments as well – with the caveat that the failure of UnixGrad to converge is not inconsistent with the findings of [17].
>
> 3. On the $\alpha^t$-weighting technique. The $\alpha^t$-weighted regret-like measure is only the first step in the analysis: the arguments involved in the convergence rate analysis of AdaWeight are otherwise completely different than in the paper of Wang & Abernethy (since there is no equivalent of our summability results for the method’s adaptive step-size). We will make sure to cite the Wang & Abernethy paper to point the reader towards other uses of the $\alpha^t$-weighting technique and to highlight the above difference.
>
>
> 4. On the use of weight-pushing for DAGs. The “dynamic programming” implementation of weight-pushing indeed requires a DAG structure. Otherwise, the corresponding path-kernel of Takimoto & Warmuth requires solving an eigenvalue problem which introduces an $O(r)$ multiplicative constant in runtime, where $r$ is the number of edges whose removal from the graph leads to a DAG. We will include this clarification, thanks for pointing it out.
>
> 5. On achieving a smooth trade-off between random and static environments. Great question, there are several points of note here. First, if we let $\mathrm{var} = \max\{\mathbb{E}[\|U_t\|^2],\mathbb{E}[\|\bar U_t\|^2]\}$ denote the variance of the observed/sampled costs (so $\mathrm{var} = 0$ in the static regime by definition), Lines 795-805 in the proof of Theorem 3 actually yield the bound
>
> $$\mathbb{E}[F(X^T) - F(x^\ast)] \leq \frac{2M_{\textrm{tot}} \log \left(\frac{P M_{\textrm{max}}}{M_{\textrm{tot}}} \right) + 16 \beta (N M_{\textrm{max}})^2 A^{\frac{3}{2}} }{ T^2} + \frac{2 \sqrt{2} \sqrt{\mathrm{var}} N M_{\textrm{max}} A }{\sqrt{T}}$$
>
> We did not include this precise expression in the paper to allow the presentation to flow faster, but we will be very happy to transfer this more precise result to the paper in order to highlight the smooth transition between the static and stochastic regimes.
>
> Furthermore, regarding the parametrization you suggest, if cost observations become more accurate over time – formally, if $\mathrm{var} = \mathcal{O}(1/T^\rho)$ for some suitable $\rho$, AdaWeight’s rate of convergence indeed carries a dependence of the order of $\mathcal{O}(T^{\mathrm{max} (-1/2 - \rho/2, - 2 )})$. We will make sure to include this remark and the relevant expressions – thanks again for your input on this.
>
>
> 6. On time-varying probability distributions. If the probability law $\mathbb{P}$ changes over time in a non-stationary way, there is no mean equilibrium flow to target. Because of this, this framework would need to be analyzed either in a dynamic regret context or in a Markovian context (if the changes of $\mathbb{P}$ follow Markovian transitions). Both cases however would lead to a completely different formulation which, while interesting, lie beyond the scope of our work. We will make sure to mention these extensions as possible directions for future work, as we believe that these would also be of interest to the community at large.
>
> 7. On minor comments. We will take care of both – namely citing the potential improvements in the deterministic case under strong convexity (the 2020 paper by Nakamura et al.), as well as the uniformization of the $\mathcal{O}$ notation for cases where it includes logarithmic terms.
>
>
>
> We trust and hope that the above alleviates your concerns, and we look forward to engaging in an open-minded discussion if you have any more questions about our paper.
>
> Kind regards,
>
> The authors of Paper 9525

---

### Official Review · Reviewer_M6uQ · 2021-07-14

**Rating:** 5
**Confidence:** 3

**Summary:**

The paper considers the speed on learning to route an equilibrium for in classical network routing models with congestion. The main application could be routing suggestions made by an application such as google maps.

They consider two versions:

(1) static: when the congestion function is fixed throughout the learning process, called static process. In this case, equilibrium is the minimizer of the potential function, so algorithms minimizing the potential can lead to fast convergence to equilibrium. This leads to a $O(1/T^2)$ error rate as a function of the time.

(2) which is called stochastic, but appears to be worse case instead, where routers can use multiplicative weight, leading to a $O(1/\sqrt{T})$ error rate.

Under some smoothness assumption on the congestion functions (bounded and Lipschitz) the paper proposes a "best-of-both worlds" algorithm called ADAWeight, that claims to achieve the $O(1/T^2)$ regret in the static case and the $O(1/\sqrt{T})$ regret in case (2).

**Limitations And Societal Impact:**

I  don't think the proposed work has negative societal impact

**Main Review:**

The algorithm proposed appears to adapt ideas from the paper [11] in ICML 2019 to achieve the claimed "best of both worlds" bounds. I wonder if this would actually give something better: a smooth tradeoff between the worst case and the static case. Something bounding the speed depending on the variation in the latency function with $O(1/T^2)$ with no variability and a maximum of $O(1/\sqrt{T})$ with variability, with a smooth transition between the two. Is such a bound possible?

For proving the best of both worlds bound a very short (1/2 a page) proof sketch is provided, which appears to be inaccessible without knowing the details of the paper [11].

I also wonder if the worst-case bound is the best way to think about a stochastic environment. If the latency function is stochastic: is the assumption that the each iteration is polled independently from a distribution? is better bound possible if that is the case?

**Time Spent Reviewing:**

1 hour

---

> ### Author Response · Authors · 2021-08-10
> **Reply to Reviewer M6uQ**
>
> Dear reviewer,
>
> Thank you for your constructive comments and positive evaluation! We provide a series of point-to-point replies to your comments below:
>
> 1. On the relation with Cutkosky (2019) [11]. We would like to stress here that the only shared element between our work and [11] is the inequality presented in Line 319. This, however, is a straightforward template inequality which only serves as a starting point for the analysis of adaptive algorithms, and has been used as such in other papers as well – e.g. [17]. The key technical contributions and the involved part of our analysis comes after this point, in establishing an upper bound for the key term $\hat\Delta_T(x^\ast)$ that appears in the RHS of this inequality (cf. Line 317) and managing the residual terms in the definition of $\eta^t$ to make sure that the result is actually summable. It is precisely this mechanism that is key for acceleration and, in this regard, there is no overlap in techniques or results with [11].
>
>
> 2. On the extent of the proof sketch provided. Sure thing, we will be happy to use the extra page to transfer the formal statements of the intermediate lemmas that we provide in the appendix, and which give a more structured and detailed outline of the proof.
>
>
> 3. On achieving a smooth trade-off between the static and stochastic regimes. Thanks for this very insightful comment, this is indeed possible. If we let $\mathrm{var} = \max\{\mathbb{E}[\|U_t\|^2],\mathbb{E}[\|\bar U_t\|^2]\}$ denote the variance of the observed/sampled costs (so $\mathrm{var} = 0$ in the static regime by definition), Lines 795-805 in the proof of Theorem 3 actually yield the bound
> $$\mathbb{E}[F(X^T) - F(x^\ast)] \leq \frac{2M_{\textrm{tot}} \log \left(\frac{P M_{\textrm{max}}}{M_{\textrm{tot}}} \right) + 16 \beta (N M_{\textrm{max}})^2 A^{\frac{3}{2}} }{ T^2} + \frac{2 \sqrt{2} \sqrt{\mathrm{var}} N M_{\textrm{max}} A }{\sqrt{T}}$$
>
> We did not include this precise expression in the paper to allow the presentation to flow faster, but we will be very happy to transfer this more precise result to the paper. Thanks again for pointing it out!
>
>
>
> 4. On the tightness of the stochastic bounds. The $\mathcal{O}(1/\sqrt{T})$ rate is, in general, unimprovable in stochastic environments, see e.g., the paper by Abernethy et al. (COLT 2008) where this is shown even in the Gaussian case (the textbooks by Nemirovski & Yudin, 1983, and Nesterov, 2004, elaborate further on this, as does the monograph of Bubeck, 2015). In this regard, convex optimization is “harder” than stochastic multi-armed bandits where logarithmic regret is achievable.
>
>
> We thank the reviewer again for their constructive comments and positive evaluation, and we’ll be happy to include the above in our revision.
>
> Kind regards,
>
> The authors of Paper 9525

---

### Official Review · Reviewer_GCju · 2021-07-16

**Rating:** 6
**Confidence:** 2

**Summary:**

The paper analyzes the Adaptive Exponential Weight Updates (AdaWeight) in the learning algorithm for dynamic path recommendations in networks. It shows that it reaches Wardrop Equilibrium at a rate that is a polynomial in the number of paths of the network and the number of epochs. The main contributions of this paper are to display that this algorithm reaches Wardrop Equilibrium in static and stochastic networks without the use of an optimizer.


**Limitations And Societal Impact:**

The problem that the paper attempts to solve seems novel.
The authors seem state the limitations about their claims and their contributions.
By providing a clear and concise introduction to the problem and a possible solution to it, the authors open up space and scope for future research.


**Main Review:**

The paper provides a detailed theoretical and experimental explanation. The implementation of AdaWeight seems new as it constructs a learning algorithm for both static and stochastic networks. Past approaches have only targeted this for either static or stochastic networks since doing so for both network seems expensive or impractical.

The paper provides sufficiently sound related work review before proceeding with explaining the background of their work. It seems clear to read and underlines the significance of the framework in the form of a time-optimized network path recommendation algorithm that can learn the network's Wardrop Equilibrium at a rate polynomial in the size of the network and the number of epochs.



**Time Spent Reviewing:**

20

---

> ### Author Response · Authors · 2021-08-10
> **Reply to Reviewer GCju**
>
> Dear reviewer,
>
> Thank you for your constructive comments and your positive evaluation! We were especially happy that you found our approach novel and paving the way to future research on the topic.
>
>
> Kind regards,
>
> The authors of Paper 9525

---

### Official Review · Reviewer_GAfB · 2021-07-18

**Rating:** 6
**Confidence:** 3

**Summary:**

Regret-minimizing routing in nonatomic congestion games is a well-studied topic and it is known that simple, classical algorithms, such as multiplicative weights, fare well in this context. Specifically, such algorithms can converge to an equilibrium at a rate of O(1/√T) when cost functions are noisy. However, this leaves something to be desired; when the environment is static (no perturbations to cost functions are possible), this convergence rate is not optimal, motivating the search for algorithms with faster convergence rates. Such algorithms do exist but might not be robust when perturbations do occur. The key contribution of the paper is an adaptive exponential weights algorithm (ADAWEIGHT) that effectively interpolates between the two algorithmic approaches. An appealing property of ADAWEIGHT is that it does not require knowing the specifics of the environment and is scalable (in the size of the network).

**Limitations And Societal Impact:**

I see no possible negative societal impact.

**Main Review:**

The paper tackles a classical problem (routing in nonatomic congestion games) and presents an interesting perspective on this problem.

While the approach is fairly incremental (and interpolates between two well-understood algorithmic schemes), I found the contributions and, in particular, the manner in which these two approaches were welded, interesting.

**Time Spent Reviewing:**

1.5

---

> ### Author Response · Authors · 2021-08-10
> **Reply to Reviewer GAfB**
>
> Dear reviewer,
>
> First of all, thank you for your thoughtful input and positive evaluation!
>
> Regarding the relation between AdaWeight and the non-adaptive optimal algorithms (AcceleWeight and ExpWeight), we would only like to point out that AdaWeight is not a “convex combination” of the two algorithms: it requires the introduction of a new primal-dual averaging technique and two “extra-gradient” steps that are altogether absent from either of the non-adaptive algorithms. It is for this reason that we include in our analysis several new techniques to bound the key term $\hat\Delta_t(x^\ast)$ (cf. Line 317) and a completely new way to treat the learning rate $\eta^t$ in order to make these bounds summable (cf. Line 324). Given the lack of a filtration-dependent step-size, these issues do not arise in any of these two well-understood non-adaptive  algorithmic schemes, hence the highly increased degree of difficulty in the analysis of AdaWeight.
>
> We will of course be happy to include this remark when we introduce AdaWeight to make the distinction sharper from ExpWeight and AcceleWeight.
>
> Thank you again for your input and positive assessment!
>
> Best regards,
>
> The authors of Paper 9525

---

### Decision · Program_Chairs · 2021-09-27

**Decision:**

Accept (Poster)

**Comment:**

This work studies the question of learning an equilibrium in routing games and provides a  smooth interpolation between static results with O(1/T^2) convergence and worst-case results with O(1/\sqrt{T}) convergence rates. This is a relatively narrow contribution, but the reviewers found the results interesting and technically non-trivial.